# *LOXL1* gene polymorphisms are associated with exfoliation syndrome/exfoliation glaucoma risk: An updated meta-analysis

Xiaoyan Li[1], Jie He[2][¤]*, Jian Sun[2]

**1** Department of Endocrinology, Clinical Medical College and The First Affiliated Hospital of Chengdu Medical College, Chengdu, Sichuan, China, **2** Department of Pulmonary and Critical Care Medicine, Clinical Medical College and The First Affiliated Hospital of Chengdu Medical College, Chengdu, Sichuan, China

¤ Current address: Chengdu, Sichuan, China
* 13540246974@163.com

**Data Availability Statement:** All relevant data are within the paper and its Supporting information files.

## Abstract

### Background

Single nucleotide polymorphisms (SNPs) in the gene encoding LOXL1 are risk factors for exfoliation syndrome and exfoliation glaucoma. This meta-analysis comprehensively investigated the association between *LOXL1* gene polymorphisms (rs1048661, rs3825942, and rs2165241) and the risk of exfoliation syndrome/exfoliation glaucoma (XFS)/(XFG).

### Methods

All eligible case-control studies, published before August 17, 2020, were searched on Medline (Ovid), PubMed, CNKI, EMBASE, and Wanfang databases.

### Results

In total, 5022 cases and 8962 controls were included in this meta-analysis. Significant associations between *LOXL1* gene polymorphisms and XFS/XFG risk was observed in the disease types-based subgroups. In addition, in the subgroup analysis of ethnicity, positive associations between *LOXL1* gene polymorphisms (rs1048661, rs3825942, and rs2165241) and XFS/XFG risk were found in Caucasians. Furthermore, rs1048661 and rs3825942 polymorphisms were related to XFS/ XFG risk in Asians; however, no significant association was observed between the *LOXL1* gene rs2165241 polymorphism and XFS/ XFG risk in Asians. In addition, rs1048661 and rs3825942 correlated with XFS/XFG susceptibility in Africans.

### Conclusions

Our results implicate *LOXL1* gene polymorphisms as XFS/XFG risk factors, especially in Caucasians.

**Funding:** The authors received no specific funding for this work.

**Competing interests:** There are no conflicts of interest to declare.

## Introduction

Exfoliation syndrome (XFS) is an age-related, generalized disorder of the extracellular matrix characterized by progressive accumulation of abnormal fibrillar material in intra- and extra-ocular tissues [1, 2]. It is estimated to affect around 80 million people worldwide, and 10–20% of people aged >60 years are severely affected by XFS [3, 4]. This disorder is also associated with a progressive form of chronic open-angle glaucoma [2] and is the second most common cause of irreversible blindness globally.

Exfoliation glaucoma (XFG) is the most common form of secondary open-angle glaucoma and occurs in the context of XFS [4, 5]. Approximately 44% of XFS cases are estimated to progress to XFG [6]. XFG is characterized by deposition of exfoliation material in the anterior segment of the eyes, obstructing aqueous humor outflow, resulting in elevated intraocular pressure and secondary open-angle glaucoma [7]. Relative to primary open-angle glaucoma, XFS-associated secondary open-angle glaucoma is associated with a more severe prognosis, higher elevated intraocular pressure, and more severe optic nerve lesions at the time of diagnosis [8]. However, the mechanism of exfoliation material production is unclear.

XFS/XFG is a multifactorial disease involving a complex interaction between numerous risk factors, including genetic and environmental factors, myopia, cigarette smoking, and diabetes [9]. XFS/XFG prevalence varies widely across populations and geographical regions, ranging from <0.4% to >20% [10]. Thus, recent studies have increasingly focused on the relationship between gene polymorphisms and XFS/XFG susceptibility.

The lysyl oxidase-like 1 (*LOXL1*) gene has been extensively studied [11–14]. The *LOXL* family comprises five genes (*LOXL, LOXL1, LOXL2, LOXL3* and *LOXL4*), which encode enzymes involved in fibrillin, elastin, and collagen cross-linking reactions (2). LOXL1, which catalyzes the oxidative deamination of tropoelastin lysine residues, is essential for elastogenesis [15]. A 2007 genome-wide association study of a Scandinavian population, found a significant association between XFS/XFG and the *LOXL1* single nucleotide polymorphisms, rs1048661, rs3825942, and rs2165241, located on chromosome 15q24.1 [16]. Since then, numerous studies have affirmed that the *LOX1* polymorphisms are associated with XFS/XFG in various populations, including Caucasians, Latin Americans, Africans and Asians. Dubey et al. [17] reported that *LOXL1* G (rs1048661), G (rs3825942) and T (rs2165241) alleles are XFS/XFG risk factors in Asians, which were similar to the results found in the original study conducted by Thorleifsson et al (2007) in Caucasians (Scandinavian population), as well as in most studies carried out in Caucasians [13, 16]. However, in most studies in Asians, the alleles T and C of rs1048661 and rs2165241, respectively, are the risk alleles. Tanito et al. [18], Ozaki et al. [19], Fuse et al. [20] and Hayashi et al. [21] reported that the alleles T of rs1048661 as well as the alleles C of rs2165241 are associated with increased risk of XFS/XFG in the Japanese population. Park DY et al. [22] and Sagong et al. [11] also found a similar phenomenon in Koreans. Similar observations were made by Chen et al. in Chinese [23]. Moreover, De Juan-Marcos et al. [24] showed that the G allele of rs3825942 and the T allele of rs2165241 were XFS/XFG risk factors in a Spanish population. However, in contrast to what was observed in most Caucasian populations, no significant association between XFS/XFG and SNP rs1048661 was observed. In addition, Rautenbach et al. [25] and Williams et al. [26] indicated that the G allele of rs3825942 was protective in Black South Africans, and the G allele of rs1048661 was a risk allele for XFS/XFG. Therefore, the associations of *LOXL1* gene polymorphisms (rs1048661, rs2165241, rs3825942) may differ across patients of different ethnicities.

Despite the existence of discrepancies between some studies related to the risk alleles of *LOXL1* SNPs, it is widely accepted that *LOXL1* gene is the most important genetic risk factor known so far for XFS/XFG. Additionally, a single study may be insufficient to explore the

small effect of *LOXL1* gene polymorphisms on XFS/XFG susceptibility, especially when the sample size is small. Given the associations between *LOXL1* gene polymorphisms and XFS/XFG pathogenesis, we carried out an updated meta-analysis on the correlation between *LOXL1* gene polymorphisms (rs1048661, rs2165241, rs3825942) and XFS/XFG risk. To our knowledge, this is the most comprehensive and accurate meta-analysis of *LOXL1* gene polymorphisms in the context of XFS/XFG susceptibility.

## Materials and methods

### Search strategy and criteria

Medline (Ovid), PubMed, CNKI, EMBASE, and Wanfang database searches for articles published before August 17, 2020, were performed using the following terms: "Lysyl oxidase-like 1", "*LOXL1*", "Exfoliation syndrome", "XFS", "Exfoliation glaucoma", "XFG", and "Polymorphism". Articles were included if: 1) they examined the relationship between XFS/XFG susceptibility and *LOXL1* variations, 2) they were case-control studies, and 3) they had complete genotype frequency data. Articles were excluded if: 1) they lacked a control group, 2) the presented data was incomplete, 3) they were duplicate publications, and 4) controls failed to meet Hardy Weinberg Equilibrium (HWE) standards.

### Quality score evaluation

The quality of the included studies was determined using the Newcastle-Ottawa Scale [27] which assesses quality based on selection, comparability, and exposure in the study. Quality scores ranged from 0 to 9. Studies scoring >6 were considered high quality. Furthermore, study quality was determined by consensus between authors.

### Data extraction

Two independent investigators extracted tangible data from each study based on the inclusion criteria. In the case of divergent views, a third author examined the controversial articles. For each study, the first author, country, publication year, ethnicities, sample size, genotyping method, and genotype frequency in the case and control groups, were extracted.

### Statistical analyses

All analyses were conducted using STATA 10.0 and RevMan 5.2. The Odds ratio (OR) and 95% confidence interval (CI) were used to estimate the association between the *LOXL1* gene polymorphisms and XFS/XFG susceptibility. Heterogeneity among studies was evaluated using the $\chi^2$-based Q statistic and a *p value* $\leq 0.1$ was considered statistically significant. When the *p* value was >0.1, the pooled OR of each study was calculated using a fixed-effects model. Otherwise, a random-effects model was used. The significance of the pooled OR was demonstrated using the Z-test and a *p value* $\leq 0.05$ was considered statistically significant. The association between *LOXL1* gene polymorphisms and XFS/XFG risk was evaluated in different genetic models. To assess the effects of ethnicity and disease type, we performed additional subgroup analyses based on ethnicity and disease type. Sensitivity analysis was carried out to assess the stability of the results. Hardy Weinberg equilibrium was evaluated using Pearson's $\chi^2$ test, and $p \geq 0.05$ was considered statistically significant.

### Publication bias

Publication bias was determined using asymmetry Begger's plots and Egger's tests [28, 29] and was carried out using STATA 10.0.

## Results

### Study characteristics

Our initial literature search returned 197 articles. Upon browsing the titles and abstracts, 111 articles were excluded, leaving 86 articles that underwent full-text review. Of the 86 articles, 41 articles were excluded because 32 articles involved other *LOXL1* gene polymorphisms (rs4461027, rs4886761, and rsl6958477), and six articles were excluded because they were not case-control studies, and three articles were excluded for meta-analyses. Then the remaining 45 full-text articles were assessed for eligibility, although five articles [10, 30–33] had been analyzed in a previous meta-analysis [34], we excluded them because three articles [10, 30, 31] did not achieve HWE in the control group, and two articles [32, 33] reported the relationship between *LOXL1* polymorphisms and primary open-angle glaucoma. This process yielded 40 case-control articles [9, 11–14, 16–26, 35–58] that were eligible for our study (Table 1). Of these, 38 articles [9, 11–14, 16–26, 35–56] studied rs1048661, 22 articles [11, 17–20, 22–24, 35, 38–42, 45–47, 49, 52, 55–57] involved rs2165241, and 38 articles [9, 11–14, 16–26, 35–37, 39–55, 57, 58] involved rs3825942 (Fig 1).

### Quantitative synthesis of data

**rs1048661 *LOXL1* gene polymorphism.** Thirty-eight articles that examined the relationship between the *LOXL1* gene polymorphism, rs1048661, and XFS/XFG risk were included in this meta-analysis. Some studies recruited XFS and XFG patients as research subjects, but these subjects did not distinguish XFS patients from XFG patients when DNA samples were sequenced. Thus, in the subgroup analysis based on the type of disease, we only extracted data from studies in which disease types (XFS or XFG) are clearly illustrated. In the subgroup analysis based on ethnicity, we combined all types of studies (XFS, XFG, XFS/XFG) to conduct the meta-analysis. Because the reason that analysis of SNPs by ethnicity is more comprehensive, we choose its merger result as the overall result. Although negative associations were found in the total sample (G vs. T, OR:1.13,95%CI: 0.85–1.52, $p$:0.40), allelic contrast analysis revealed positive associations in the XFS (G vs. T, OR: 1.50,95%CI: 1.16–1.93, $p<0.001$) and XFG (G vs. T, OR: 1.97,95%CI: 1.45–2.66, $p<0.001$) subgroups. (Fig 2, Table 2). The rs1048661 G allele was significantly correlated with higher XFG and XFS risk relative to the T allele. In the subgroup analysis of ethnicity, the meta-analysis indicated a significant association between the *LOXL1* polymorphism (rs1048661) and XFS/XFG risk in Africans (G vs. T, OR: 23.42, 95%CI: 4.48–122.47, $p < 0.001$) (Fig 3, Table 2). Notably, allelic contrast analysis showed that XFS/XFG susceptibility markedly increased in Caucasians (G vs. T, OR:1.99, 95%CI: 1.70–2.33, $p <0.001$) and significantly decreased in Asians (G vs. T, OR: 0.52, 95%CI: 0.29–0.94, $p$:0.03) (Fig 3, Table 2). In Asians, the association between rs1048661 alleles and risk was opposite to that in Caucasians and Africans. A summary of the results from other comparative genetic models is shown in Table 2.

**rs2165241 *LOXL1* gene polymorphism.** Twenty-two case-control articles on the relationship between the *LOXL1* gene polymorphism, rs2165241, and XFS/XFG risk were included in the meta-analysis. Overall analyses revealed a significant association between XFS/XFG susceptibility and the rs2165241 (T vs. C, OR: 1.61, 95%CI: 1.18–2.19, $p$:0.002) polymorphism (Table 2). The results revealed that genetic polymorphism of *LOXL1*(rs2165241) was associated with susceptibility to XFS (T vs. C, OR: 2.14, 95%CI: 1.33–3.45, $p$:0.002) and XFG (T vs. C, OR: 2.00, 95%CI: 1.21–3.31, $p$:0.007) (Fig 4, Table 2), in the allelic contrast.

Subgroup analysis by ethnicity identified an increased risk in Caucasians (T vs. C, OR: 2.76, 95%CI: 1.99–3.84, $p <0.001$) (Fig 5, Table 2). However, there was no significant association

**Table 1. Characteristics of case-control studies included in meta-analysis on LOXL1 gene polymorphism (rs1048661, rs2165241, rs3825942).**

| First author | Year | Origin | Ethnicity | Type | Case | Control | Case | | | Control | | | NOS | Hardy-Weinberg equilibrium |
|---|---|---|---|---|---|---|---|---|---|---|---|---|---|---|
| **Ref No rs1048661** | | | | | | | GG | GT | TT | GG | GT | TT | | |
| Lan [35] | 2020 | China | Asian | XFG | 91 | 180 | 57 | 32 | 2 | 49 | 90 | 41 | 7 | YES |
| Taghavi [36] | 2019 | Iran | Asian | XFS | 60 | 40 | 48 | 12 | 0 | 24 | 16 | 0 | 7 | YES |
| Pandav [37] | 2018 | India | Asian | XFG | 30 | 61 | 17 | 10 | 3 | 41 | 16 | 4 | 6 | YES |
| Pandav [37] | 2018 | India | Asian | XFS | 27 | 61 | 15 | 10 | 2 | 41 | 16 | 4 | 6 | YES |
| Shihadeh [38] | 2018 | Jordan | Asian | XFS/XFG | 61 | 59 | 46 | 15 | 0 | 44 | 14 | 1 | 6 | YES |
| Yaz [39] | 2018 | Turkey | Caucasian | XFG | 58 | 171 | 46 | 12 | 0 | 87 | 64 | 20 | 6 | YES |
| Yaz [39] | 2018 | Turkey | Caucasian | XFS | 58 | 171 | 32 | 26 | 0 | 87 | 64 | 20 | 6 | YES |
| Asfuroglu [40] | 2017 | Turkey | Caucasian | XFS | 44 | 47 | 17 | 27 | 0 | 25 | 21 | 1 | 7 | YES |
| Asfuroglu[40] | 2017 | Turkey | Caucasian | XFG | 65 | 47 | 14 | 50 | 1 | 25 | 21 | 1 | 7 | YES |
| De Juan-Marcos [24] | 2016 | Spain | Caucasian | XFS | 60 | 90 | 33 | 25 | 2 | 47 | 35 | 8 | 6 | YES |
| De Juan-Marcos [24] | 2016 | Spain | Caucasian | XFG | 40 | 90 | 24 | 16 | 0 | 47 | 35 | 8 | 6 | YES |
| Gayathri [13] | 2016 | Germany and Italy | Caucasian | XFS | 48 | 40 | 26 | 20 | 2 | 15 | 20 | 5 | 8 | YES |
| Alvarez [41] | 2015 | Spain | Caucasian | XFG | 105 | 200 | 75 | 27 | 3 | 80 | 94 | 26 | 6 | YES |
| Qiu [42] | 2015 | China | Asian | XFS | 152 | 228 | 106 | 42 | 4 | 109 | 98 | 21 | 7 | YES |
| Dubey [17] | 2014 | Indian | Asian | XFS | 150 | 225 | 93 | 46 | 11 | 108 | 91 | 26 | 7 | YES |
| Dubey [17] | 2014 | Indian | Asian | XFG | 150 | 225 | 102 | 40 | 8 | 108 | 91 | 26 | 7 | YES |
| Anastasopoulos [43] | 2014 | Greece | Caucasian | XFS | 40 | 93 | 24 | 15 | 1 | 51 | 39 | 3 | 7 | YES |
| Anastasopoulos [43] | 2014 | Greece | Caucasian | XFG | 34 | 93 | 24 | 10 | 0 | 51 | 39 | 3 | 7 | YES |
| Chiras [14] | 2013 | Greece | Caucasian | XFS | 54 | 93 | 33 | 19 | 2 | 49 | 39 | 5 | 6 | YES |
| Chiras [14] | 2013 | Greece | Caucasian | XFG | 70 | 93 | 56 | 13 | 1 | 49 | 39 | 5 | 6 | YES |
| Kasim [9] | 2013 | Turkey | Caucasian | XFS | 100 | 100 | 77 | 22 | 1 | 52 | 38 | 10 | 6 | YES |
| Kasim [9] | 2013 | Turkey | Caucasian | XFG | 100 | 100 | 74 | 26 | 0 | 52 | 38 | 10 | 6 | YES |
| Park [22] | 2013 | Korea | Asian | XFS/XFG | 110 | 127 | 1 | 4 | 105 | 13 | 49 | 65 | 7 | YES |
| Michael [44] | 2012 | Pakistan | Asian | XFG | 128 | 180 | 91 | 36 | 1 | 78 | 81 | 21 | 6 | YES |
| Rautenbach [25] | 2011 | South African | African | XFS | 43 | 47 | 43 | 0 | 0 | 37 | 9 | 1 | 7 | YES |
| Mayinu [45] | 2011 | China | Asian | XFS/XFG | 64 | 127 | 42 | 20 | 2 | 60 | 56 | 11 | 7 | YES |
| Malukiewicz [46] | 2011 | Poland | Caucasian | XFS | 36 | 30 | 29 | 7 | 0 | 20 | 8 | 2 | 6 | YES |
| Sagong [11] | 2011 | Korea | Asian | XFS | 28 | 146 | 0 | 4 | 24 | 22 | 60 | 64 | 6 | YES |
| Sagong [11] | 2011 | Korea | Asian | XFG | 61 | 146 | 4 | 1 | 56 | 22 | 60 | 64 | 6 | YES |
| Williams [26] | 2010 | South African | African | XFG | 50 | 50 | 49 | 1 | 0 | 33 | 15 | 2 | 7 | YES |
| Wolf [47] | 2010 | German | Caucasian | XFG | 128 | 266 | 89 | 38 | 1 | 110 | 131 | 25 | 8 | YES |
| Abu-Amero [48] | 2010 | Saudi Arabia | Asian | XFG | 93 | 101 | 72 | 19 | 2 | 57 | 40 | 4 | 6 | YES |
| Chen [23] | 2009 | China | Asian | XFS/XFG | 50 | 125 | 4 | 3 | 43 | 23 | 75 | 27 | 5 | YES |
| Lemmela [49] | 2009 | Finland | Caucasian | XFS/XFG | 126 | 325 | 88 | 32 | 6 | 152 | 140 | 33 | 7 | YES |
| Lee [50] | 2009 | China | Asian | XFS/XFG | 62 | 171 | 20 | 25 | 17 | 29 | 94 | 48 | 6 | YES |
| Ozaki [19] | 2008 | Japan | Asian | XFS/XFG | 209 | 172 | 2 | 18 | 189 | 45 | 81 | 46 | 6 | YES |
| Hewitt [51] | 2008 | America | Caucasian | XFS | 86 | 2087 | 56 | 22 | 8 | 904 | 947 | 236 | 8 | YES |
| Challa [52] | 2008 | America | Caucasian | XFG | 47 | 215 | 29 | 16 | 2 | 99 | 88 | 28 | 7 | YES |
| Fuse [20] | 2008 | Japan | Asian | XFS/XFG | 56 | 138 | 1 | 2 | 53 | 28 | 80 | 30 | 7 | YES |
| Mabuchi [53] | 2008 | Japan | Asian | XFS/XFG | 302 | 191 | 47 | 108 | 147 | 40 | 92 | 59 | 6 | YES |
| Mossbock [54] | 2008 | Australia | Caucasian | XFG | 167 | 170 | 119 | 43 | 5 | 79 | 70 | 21 | 6 | YES |
| Aragon-Martin [55] | 2008 | America | Caucasian | XFS/XFG | 283 | 330 | 197 | 83 | 3 | 162 | 140 | 28 | 7 | YES |
| Pasutto [56] | 2008 | Germany and Italy | Caucasian | XFS | 280 | 412 | 179 | 91 | 10 | 170 | 194 | 48 | 7 | YES |
| Pasutto [56] | 2008 | Germany and Italy | Caucasian | XFG | 441 | 412 | 302 | 130 | 9 | 170 | 194 | 48 | 7 | YES |
| Ramprasad [12] | 2008 | Indian | Asian | XFS/XFG | 52 | 97 | 29 | 17 | 6 | 36 | 51 | 10 | 6 | YES |
| Hayashi [21] | 2008 | Japan | Asian | XFS/XFG | 59 | 189 | 0 | 1 | 58 | 37 | 100 | 52 | 7 | YES |

*(Continued)*

**Table 1.** (Continued)

| First author | Year | Origin | Ethnicity | Type | Case | Control | Case | | | Control | | | NOS | Hardy-Weinberg equilibrium |
|---|---|---|---|---|---|---|---|---|---|---|---|---|---|---|
| Tanito [18] | 2008 | Japan | Asian | XFS/XFG | 142 | 251 | 2 | 10 | 130 | 65 | 143 | 43 | 6 | YES |
| Thorleifsson [16] | 2007 | Iceland | Caucasian | XFS/XFG | 128 | 1024 | 86 | 35 | 7 | 414 | 477 | 133 | 8 | YES |
| **rs2165241** | | | | | | | TT | TC | CC | TT | TC | CC | | |
| Lan [35] | 2020 | China | Asian | XFG | 91 | 180 | 43 | 34 | 14 | 90 | 70 | 20 | 7 | YES |
| Shihadeh [38] | 2018 | Jordan | Asian | XFS/XFG | 61 | 59 | 38 | 20 | 3 | 42 | 12 | 5 | 7 | YES |
| Yaz [39] | 2018 | Turkey | Caucasian | XFS | 48 | 171 | 28 | 18 | 2 | 31 | 88 | 52 | 6 | YES |
| Yaz [39] | 2018 | Turkey | Caucasian | XFG | 58 | 171 | 37 | 18 | 3 | 31 | 88 | 52 | 6 | YES |
| Asfuroglu [40] | 2017 | Turkey | Caucasian | XFS | 44 | 47 | 21 | 23 | 0 | 9 | 31 | 7 | 7 | YES |
| Asfuroglu [40] | 2017 | Turkey | Caucasian | XFG | 64 | 47 | 39 | 25 | 0 | 9 | 31 | 7 | 7 | YES |
| De Juan-Marcos [24] | 2016 | Spain | Caucasian | XFS | 60 | 90 | 6 | 29 | 25 | 28 | 38 | 24 | 6 | YES |
| De Juan-Marcos [24] | 2016 | Spain | Caucasian | XFG | 40 | 90 | 2 | 14 | 24 | 28 | 38 | 24 | 6 | YES |
| Alvarez [41] | 2015 | Spain | Caucasian | XFG | 105 | 200 | 70 | 29 | 6 | 41 | 104 | 55 | 6 | YES |
| Qiu [42] | 2015 | China | Asian | XFS | 152 | 228 | 42 | 75 | 35 | 28 | 96 | 104 | 7 | YES |
| Dubey [17] | 2014 | Indian | Asian | XFS | 150 | 224 | 42 | 69 | 39 | 14 | 88 | 122 | 7 | YES |
| Dubey [17] | 2014 | Indian | Asian | XFG | 150 | 224 | 42 | 64 | 44 | 14 | 88 | 122 | 7 | YES |
| Park [22] | 2013 | Korea | Asian | XFS/XFG | 101 | 115 | 0 | 2 | 99 | 1 | 13 | 101 | 7 | YES |
| Mayinu [45] | 2011 | China | Asian | XFS/XFG | 64 | 127 | 22 | 28 | 14 | 10 | 42 | 75 | 7 | YES |
| Malukiewicz [46] | 2011 | Poland | Caucasian | XFS | 36 | 30 | 28 | 7 | 1 | 14 | 11 | 5 | 6 | YES |
| Sagong [11] | 2011 | Korea | Asian | XFS | 28 | 146 | 0 | 0 | 28 | 3 | 21 | 122 | 6 | YES |
| Sagong [11] | 2011 | Korea | Asian | XFG | 61 | 146 | 1 | 1 | 59 | 3 | 21 | 122 | 6 | YES |
| Wolf [47] | 2010 | German | Caucasian | XFG | 101 | 280 | 60 | 38 | 3 | 70 | 135 | 75 | 8 | YES |
| Lemmela [49] | 2009 | Finland | Caucasian | XFS/XFG | 140 | 316 | 76 | 53 | 11 | 65 | 166 | 85 | 7 | YES |
| Chen [23] | 2009 | China | Asian | XFS/XFG | 50 | 125 | 0 | 2 | 48 | 0 | 25 | 100 | 5 | YES |
| Ozaki [19] | 2008 | Japan | Asian | XFS/XFG | 209 | 172 | 2 | 3 | 204 | 3 | 29 | 140 | 6 | YES |
| Challa [52] | 2008 | America | Caucasian | XFG | 50 | 235 | 29 | 17 | 4 | 76 | 114 | 45 | 7 | YES |
| Yang [57] | 2008 | America | Caucasian | XFS/XFG | 62 | 170 | 51 | 9 | 2 | 49 | 81 | 40 | 6 | YES |
| Tanito [18] | 2008 | Japan | Asian | XFS/XFG | 142 | 251 | 0 | 2 | 140 | 5 | 47 | 199 | 6 | YES |
| Fuse [20] | 2008 | Japan | Asian | XFS/XFG | 56 | 138 | 0 | 2 | 54 | 0 | 16 | 122 | 7 | YES |
| Aragon-Martin [55] | 2008 | America | Caucasian | XFS/XFG | 284 | 328 | 149 | 119 | 16 | 60 | 174 | 94 | 7 | YES |
| Pasutto [56] | 2008 | Germany and Italy | Caucasian | XFS | 276 | 408 | 154 | 102 | 20 | 104 | 187 | 117 | 7 | YES |
| Pasutto [56] | 2008 | Germany and Italy | Caucasian | XFG | 441 | 408 | 272 | 143 | 26 | 104 | 187 | 117 | 7 | YES |
| **rs3825942** | | | | | | | GG | GA | AA | GG | GA | AA | | |
| Lan [35] | 2020 | China | Asian | XFG | 91 | 176 | 76 | 15 | 0 | 150 | 23 | 3 | 7 | YES |
| Kobakhidze [58] | 2019 | Georgia | Asian | XFS | 132 | 194 | 99 | 28 | 5 | 102 | 62 | 30 | 7 | YES |
| Taghavi [36] | 2019 | Iran | Asian | XFS | 60 | 40 | 60 | 0 | 0 | 19 | 20 | 1 | 7 | YES |
| Pandav [37] | 2018 | India | Asian | XFG | 30 | 61 | 26 | 4 | 0 | 41 | 16 | 4 | 6 | YES |
| Pandav [37] | 2018 | India | Asian | XFS | 27 | 61 | 20 | 7 | 0 | 41 | 16 | 4 | 6 | YES |
| Yaz [39] | 2018 | Turkey | Caucasian | XFG | 58 | 171 | 58 | 0 | 0 | 108 | 57 | 6 | 6 | YES |
| Yaz [39] | 2018 | Turkey | Caucasian | XFS | 48 | 171 | 48 | 0 | 0 | 108 | 57 | 6 | 6 | YES |
| Asfuroglu [40] | 2017 | Turkey | Caucasian | XFS | 44 | 47 | 26 | 10 | 8 | 44 | 3 | 0 | 7 | YES |
| Asfuroglu [40] | 2017 | Turkey | Caucasian | XFG | 65 | 47 | 53 | 7 | 5 | 44 | 3 | 0 | 7 | YES |
| De Juan-Marcos [24] | 2016 | Spain | Caucasian | XFS | 60 | 90 | 58 | 1 | 1 | 66 | 21 | 3 | 6 | YES |
| De Juan-Marcos [24] | 2016 | Spain | Caucasian | XFG | 40 | 90 | 37 | 3 | 0 | 66 | 21 | 3 | 6 | YES |
| Gayathri [13] | 2016 | Germany | Caucasian | XFS | 48 | 40 | 45 | 3 | 0 | 26 | 9 | 5 | 8 | YES |
| Álvarez [41] | 2015 | Spain | Caucasian | XFG | 105 | 200 | 103 | 2 | 0 | 144 | 50 | 6 | 6 | YES |
| Qiu [42] | 2015 | China | Asian | XFS | 152 | 228 | 140 | 10 | 2 | 147 | 77 | 4 | 7 | YES |
| Dubey [17] | 2014 | Indian | Asian | XFS | 150 | 225 | 143 | 6 | 1 | 107 | 100 | 18 | 7 | YES |

*(Continued)*

**Table 1.** (Continued)

| First author | Year | Origin | Ethnicity | Type | Case | Control | Case | | | Control | | | NOS | Hardy-Weinberg equilibrium |
|---|---|---|---|---|---|---|---|---|---|---|---|---|---|---|
| Dubey [17] | 2014 | Indian | Asian | XFG | 150 | 225 | 138 | 5 | 7 | 107 | 100 | 18 | 7 | YES |
| Anastasopoulos [43] | 2014 | Greece | Caucasian | XFS | 40 | 93 | 39 | 1 | 0 | 61 | 31 | 1 | 7 | YES |
| Anastasopoulos [43] | 2014 | Greece | Caucasian | XFG | 34 | 93 | 33 | 1 | 0 | 61 | 31 | 1 | 7 | YES |
| Chiras [14] | 2013 | Greece | Caucasian | XFS | 53 | 97 | 36 | 17 | 0 | 48 | 45 | 4 | 6 | YES |
| Chiras [14] | 2013 | Greece | Caucasian | XFG | 71 | 97 | 52 | 19 | 0 | 48 | 45 | 4 | 6 | YES |
| Kasim [9] | 2013 | Turkey | Caucasian | XFS | 100 | 100 | 100 | 0 | 0 | 71 | 26 | 3 | 6 | YES |
| Kasim [9] | 2013 | Turkey | Caucasian | XFG | 100 | 100 | 100 | 0 | 0 | 71 | 26 | 3 | 6 | YES |
| Park [22] | 2013 | Korea | Asian | XFS/XFG | 110 | 127 | 108 | 2 | 0 | 101 | 26 | 0 | 7 | YES |
| Micheal [44] | 2012 | Pakistan | Asian | XFG | 128 | 180 | 121 | 7 | 0 | 130 | 42 | 8 | 6 | YES |
| Rautenbach [25] | 2011 | South African | African | XFS | 43 | 47 | 5 | 2 | 36 | 19 | 20 | 8 | 7 | YES |
| Mayinu [45] | 2011 | China | Asian | XFS/XFG | 64 | 127 | 58 | 6 | 0 | 80 | 45 | 2 | 7 | YES |
| Malukiewicz [46] | 2011 | Poland | Caucasian | XFS | 36 | 30 | 36 | 0 | 0 | 23 | 6 | 1 | 6 | YES |
| Sagong [11] | 2011 | Korea | Asian | XFS | 28 | 146 | 27 | 1 | 0 | 116 | 27 | 3 | 6 | YES |
| Sagong [11] | 2011 | Korea | Asian | XFG | 61 | 146 | 59 | 2 | 0 | 116 | 27 | 3 | 6 | YES |
| Williams [26] | 2010 | South African | African | XFG | 50 | 50 | 2 | 9 | 39 | 20 | 22 | 8 | 7 | YES |
| Wolf [47] | 2010 | German | Caucasian | XFG | 127 | 272 | 125 | 2 | 0 | 196 | 68 | 8 | 8 | YES |
| Abu-Amero [48] | 2010 | Saudi Arabia | Asian | XFG | 93 | 101 | 88 | 4 | 1 | 70 | 25 | 6 | 6 | YES |
| Chen [23] | 2009 | China | Asian | XFS/XFG | 50 | 125 | 50 | 0 | 0 | 101 | 22 | 2 | 5 | YES |
| Lemmela [49] | 2009 | Finland | Caucasian | XFS/XFG | 126 | 325 | 119 | 6 | 1 | 224 | 87 | 14 | 7 | YES |
| Lee [50] | 2009 | China | Asian | XFS/XFG | 62 | 171 | 61 | 1 | 0 | 143 | 28 | 0 | 6 | YES |
| Ozaki [19] | 2008 | Japan | Asian | XFS/XFG | 209 | 172 | 205 | 2 | 2 | 130 | 37 | 5 | 6 | YES |
| Hewitt [51] | 2008 | America | Caucasian | XFS | 86 | 2089 | 79 | 5 | 2 | 1479 | 552 | 58 | 8 | YES |
| Challa [52] | 2008 | America | Caucasian | XFG | 50 | 235 | 45 | 5 | 0 | 177 | 51 | 7 | 7 | YES |
| Yang [57] | 2008 | America | Caucasian | XFS/XFG | 62 | 170 | 62 | 0 | 0 | 124 | 41 | 5 | 6 | YES |
| Fuse [20] | 2008 | Japan | Asian | XFS/XFG | 56 | 138 | 56 | 0 | 0 | 108 | 26 | 4 | 7 | YES |
| Mabuchi [53] | 2008 | Japan | Asian | XFS/XFG | 302 | 191 | 243 | 53 | 6 | 143 | 40 | 8 | 6 | YES |
| Mossbock [54] | 2008 | Australia | Caucasian | XFG | 167 | 170 | 165 | 2 | 0 | 109 | 60 | 1 | 6 | YES |
| Aragon-Martin [55] | 2008 | American | Caucasian | XFS/XFG | 283 | 332 | 260 | 23 | 0 | 216 | 98 | 18 | 7 | YES |
| Ramprasad [12] | 2008 | Indian | Asian | XFS/XFG | 52 | 97 | 45 | 6 | 1 | 52 | 40 | 5 | 6 | YES |
| Hayashi [21] | 2008 | Japan | Asian | XFS/XFG | 59 | 189 | 59 | 0 | 0 | 137 | 50 | 2 | 7 | YES |
| Tanito [18] | 2008 | Japan | Asian | XFS/XFG | 142 | 251 | 140 | 2 | 0 | 158 | 87 | 6 | 6 | YES |
| Thorleifsson [16] | 2007 | Iceland | Caucasian | XFS/XFG | 129 | 490 | 125 | 4 | 0 | 363 | 113 | 14 | 8 | YES |

between the *LOXL1* gene rs2165241 polymorphism and XFS/XFG risk in Asians (T vs. C, OR: 0.65, 95%CI: 0.36–1.17, *p*:0.15) (Fig 5, Table 2). A summary of the results from other comparative genetic models is also shown in Table 2.

**rs3825942 *LOXL1* gene polymorphism.** For the *LOXL1* gene polymorphism, rs3825942, 38 articles were included in our meta-analysis. Overall analyses revealed a positive *LOXL1* rs3825942 (G vs. A, OR: 5.33, 95%CI: 3.49–8.16, *p* <0.001) association with XFS/XFG susceptibility (Table 2). In the subgroup analysis by disease type, the *LOXL1* rs3825942 gene polymorphism revealed a significant association with XFS (G vs. A, OR: 4.16, 95%CI: 1.86–9.30, *p* <0.001) and XFG (G vs. A, OR: 4.72, 95%CI: 2.10–10.60, *P*<0.001) (Fig 6, Table 2) susceptibility in a genetic model, G vs. A. In subgroup analysis by ethnicity, increased risks were identified among Caucasians (G vs. A, OR: 6.48, 95%CI: 3.67–11.44, *P*<0.001) and Asians (G vs. A, OR: 5.89, 95%CI: 3.79–9.16, *p* <0.001) (Fig 7, Table 2), suggesting that variant G allele carriers are at higher risk of XFS/XFG relative to A allele carriers. In contrast, the G allele indicates

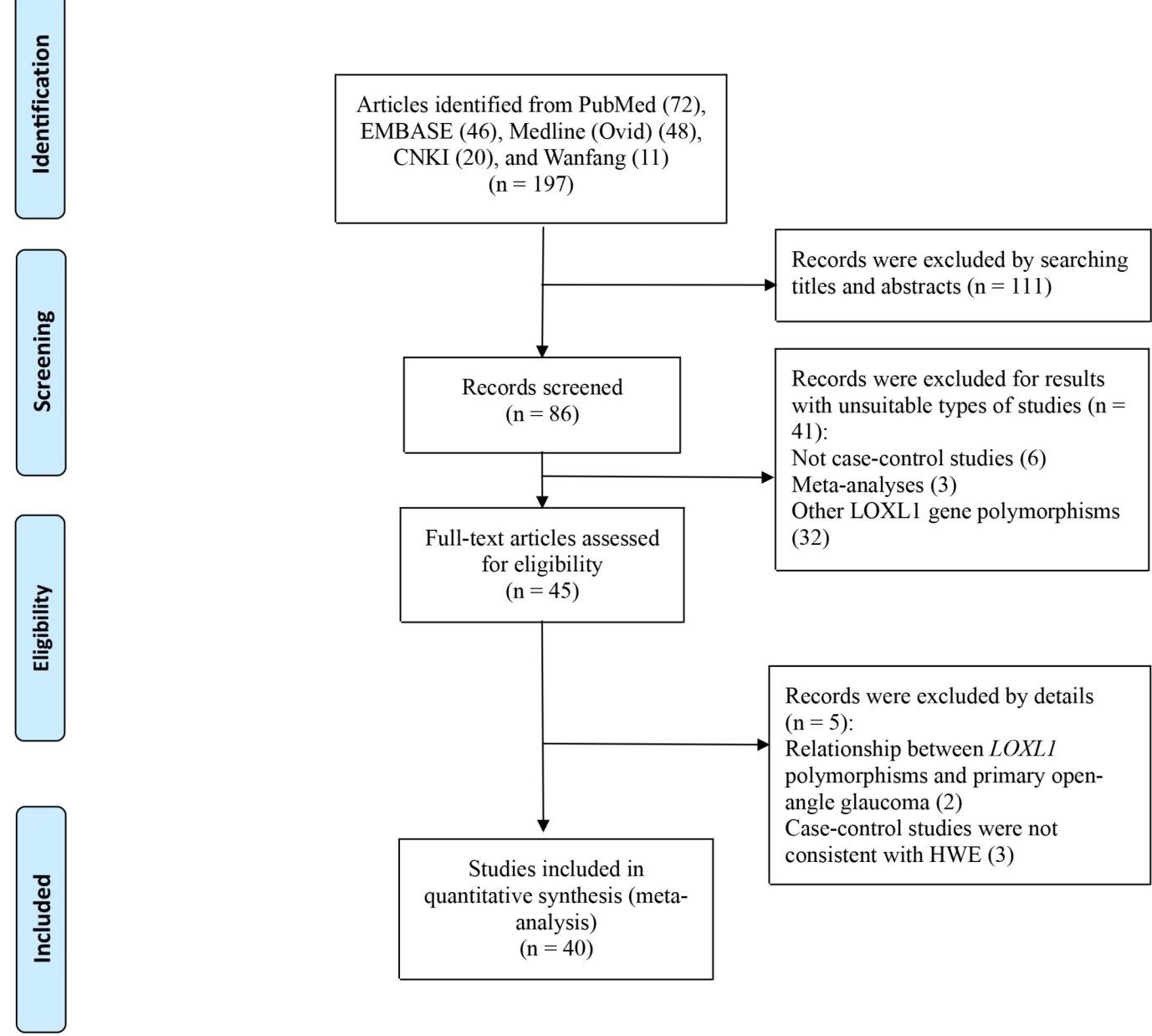

**Fig 1. Flow diagram of studies identified.**

protection from XFS/XFG in Africans (G vs. A, OR: 0.10, 95%CI: 0.06–0.16, *p* <0.001). A summary of the results from other comparative genetic models is shown in Table 2.

## Publication bias and sensitivity analyses

Funnel plot pictures were symmetrical inverted funnels. Egger's test was used to provide statistical evidence of the funnel plot (rs1048661: *t* = 1.62, *p* = 0.114; rs3825942: *t* = 1.26, *p* = 0.215; rs2165241: *t* = -2.10, *p* = 0.148) (Fig 8). To determine the potential source of heterogeneity, we performed a sensitivity analysis by sequentially excluding studies from the meta-analysis and assessing the effect of each article on the pooled results. This analysis did not reveal any significant alterations to the pooled ORs, indicating the stability of the three polymorphisms studied.

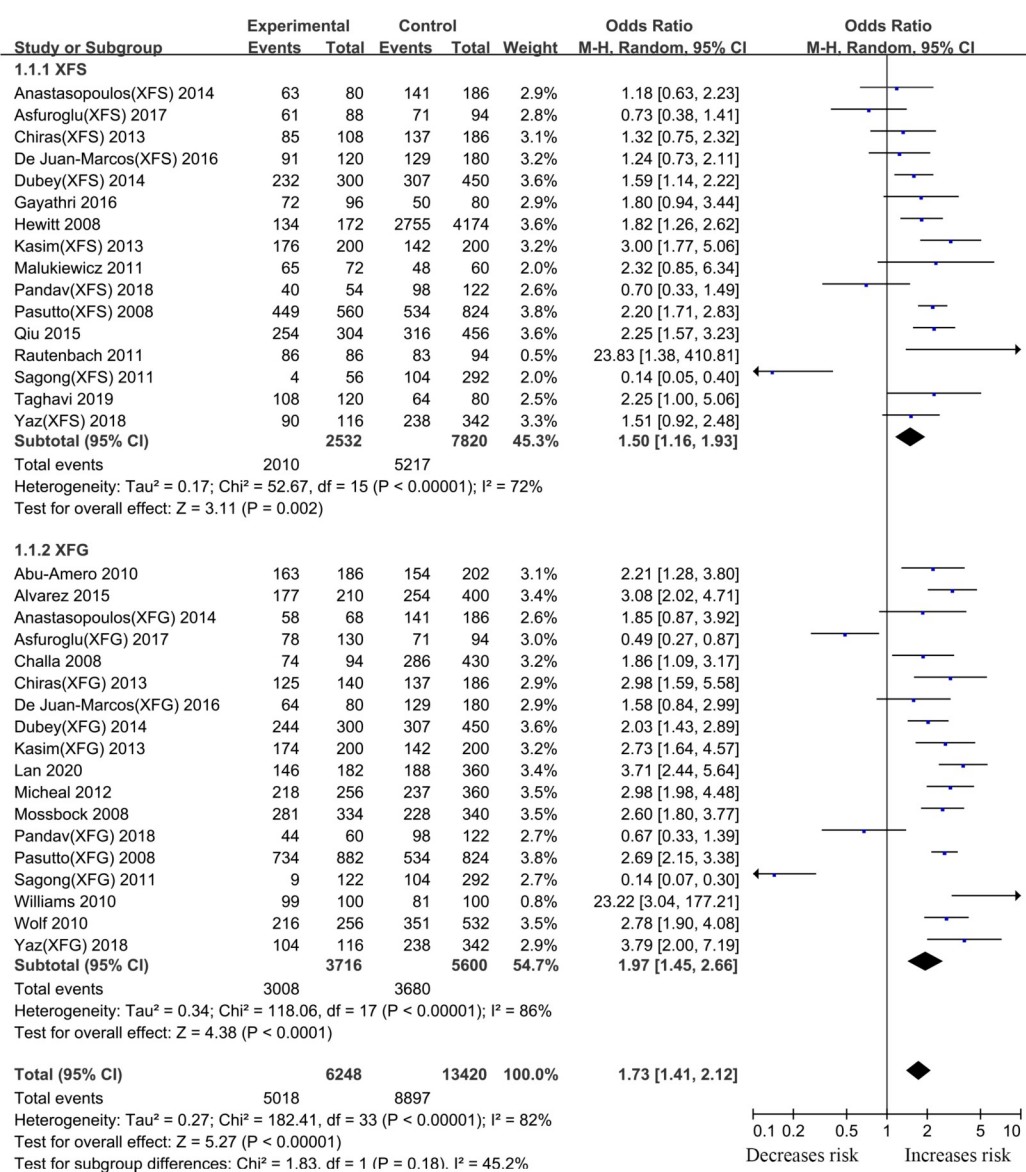

**Fig 2. Meta-analysis for the association between exfoliation syndrome/exfoliation glaucoma risks and LOXL1 gene polymorphism rs1048661 (G vs. T): Subgroup analysis by disease types (squares depict individual studies and diamonds depict summary effect size estimates (Odds Ratio, OR)).**

## Discussion

XFS is associated with a high morbidity and blindness rate [5]. This systemic disease of the extracellular matrix, may cause pathological material accumulation in blood vessels, skin, heart, lung, liver, and cerebral meninges [59]. XFG, which results from XFS, is the most common identifiable cause of secondary open-angle glaucoma and is associated with cataracts [60–62]. Additionally, XFG increases the risk of potentially sight-threatening conditions and serious complications from cataract surgery [59]. Numerous studies indicate that XFS/XFG risk factors include inflammation, immune dysfunction, oxidative stress, unhealthy lifestyle, and various environmental factors [63]. Owing to clustering of XFS/XFG in families, concordance in monozygotic twins, and prevalence variability by ethnicity, genetic factors are regarded as

**Table 2. Summary of different comparative results on *LOXL1* gene polymorphism (rs1048661, rs2165241, rs3825942).**

| Variables/SNP | Studies | Case/Control | OR (95%CI) P | OR (95%CI) P | OR (95%CI) P | OR (95%CI) P | OR (95%CI) P |
|---|---|---|---|---|---|---|---|
| **rs1048661** | | | G vs. T | GT vs.TT | GG vs TT | GG+GT vs.TT | GG vs. GT+TT |
| Total | 38 | 4828/10036 | 1.13(0.85–1.52) 0.40 | 0.98(0.57–1.67) 0.95 | 1.87(1.11–3.16) 0.019 | 1.4(0.78–2.54) 0.263 | 1.6(1.29–1.99) <0.001 |
| Ethnicity | | | | | | | |
| Asian | 19 | 2137/3240 | 0.52(0.29–0.94) 0.03 | 0.33(0.15–0.72) 0.005 | 0.54(0.23–1.27) 0.161 | 0.4(0.17–0.93) 0.033 | 0.87(0.55–1.37) 0.542 |
| Caucasian | 17 | 2598/6699 | 1.99(1.70–2.33) <0.001 | 2.22(1.69–2.92) <0.001 | 5(3.72–6.72) <0.001 | 3.59(2.73–4.71) <0.001 | 2.14(1.76–2.6) <0.001 |
| African | 2 | 93/97 | 23.42( 4.48–122.47) <0.001 | 0.48(0.02–15.52) 0.682 | 5.17(0.55–47.81) 0.148 | 3.89(0.42–35.81) 0.231 | 24.94(4.67–133.28) <0.001 |
| Disease Category | | | | | | | |
| XFS | 16 | 1266/3910 | 1.50(1.16–1.93) <0.001 | 1.44(1.06–1.95) 0.02 | 2.96(1.85–4.74) <0.001 | 2.1(1.18–3.73) 0.012 | 1.78(1.37–2.31) <0.001 |
| XFG | 18 | 1858/2800 | 1.97(1.45–2.66) <0.001 | 2.19(1.19–4.03) 0.012 | 4.81(2.41–9.6) <0.001 | 3.542(1.6–7.82) 0.002 | 2.26(1.69–3.02) <0.001 |
| **rs2165241** | | | T vs. C | TC vs.CC | TT vs.CC | TT+TC vs.CC | TT vs. TC+CC |
| Total | 22 | 3124/5126 | 1.61(1.18–2.19) 0.002 | 1.31(0.86–1.99) 0.207 | 4.47(2.59–7.7) <0.001 | 1.75(1.06–2.89) 0.028 | 2.98(.14–4.15) <0.001 |
| Ethnicity | | | | | | | |
| Asian | 11 | 1315/2135 | 0.65(0.36–1.17) 0.15 | 0.55(0.26–1.15) 0.112 | 2.11(0.93–4.79) 0.074 | 0.6(0.27–1.31) 0.198 | 1.75(0.88–3.45) 0.108 |
| Caucasian | 11 | 1809/2991 | 2.76(1.99–3.84) <0.001 | 2.48(1.6–3.85) <0.001 | 7.52(3.69–15.33) <0.001 | 4.34(2.32–8.14) <0.001 | 3.89(2.75–5.5) <0.001 |
| Disease Category | | | | | | | |
| XFS | 8 | 794/1344 | 2.14(1.33–3.45) 0.002 | 2.12(1.27–3.54) 0.004 | 4.77(1.73–13.15) 0.003 | 2.74(1.33–5.67) 0.006 | 2.82(1.55–5.11) 0.001 |
| XFG | 10 | 1161/1981 | 2(1.21–3.31) 0.007 | 1.67(1.67–3.15) 0.109 | 4.71(1.72–12.88) 0.003 | 2.44(1.05–5.67) 0.037 | 3.15(1.8–5.49) <0.001 |
| **rs3825942** | | | G vs. A | GA vs. AA | GG vs. AA | GG+GA vs. AA | GG vs.GA+ AA |
| Total | 38 | 4233/9017 | 5.33(3.49–8.16) <0.001 | 0.61(0.45–0.82) 0.001 | 2.35(1.84–2.99) <0.001 | 1.59(1.27–2) <0.001 | 6.16(4.15–9.14) <0.001 |
| Ethnicity | | | | | | | |
| Asian | 19 | 2208/3371 | 5.89(3.79–9.16) <0.001 | 1.04(0.66–1.63) 0.866 | 4.71(3.15–7.06) <0.001 | 3.41(2.28–5.1) <0.001 | 7.09(4.23–11.91) <0.001 |
| Caucasian | 17 | 1932/5549 | 6.48(3.67–11.44) <0.001 | 1.(0.55–1.82) 0.989 | 3.46(2.27–5.29) <0.001 | 2.88(1.87–4.43) <0.001 | 7.38(4.26–12.81) <0.001 |
| African | 2 | 93/97 | 0.1(0.06–0.16) <0.001 | 0.05(0.02–0.13) <0.001 | 0.04(0.01–0.1) <0.001 | 0.05(0.02–0.1) <0.001 | 0.13(0.04–0.37) <0.001 |
| Disease Category | | | | | | | |
| XFS | 17 | 1107/3698 | 4.16(1.86–9.30) <0.001 | 0.51(0.18–1.45) 0.206 | 2.61(0.96–7.1) 0.061 | 1.96(0.69–5.57) 0.205 | 5.1(2.46–10.58) <0.001 |
| XFG | 17 | 1420/2414 | 4.72(2.1–10.6) <0.001 | 0.49(0.22–1.13) 0.093 | 2.92(1.05–8.09) 0.04 | 2.36(0.84–6.69) 0.105 | 5.23(2.51–10.88) <0.001 |

XFS/XFG risk factors [64–68]. It is widely accepted that the *LOXL1* gene is the most important genetic risk factor known so far for XFS/XFG. Besides, a single study might lack sufficient power to detect the potential small *LOXL1* gene polymorphism effects associated with XFS/ XFG, especially when the sample size is not adequate. Thus, a meta-analysis may effectively identify the association between genetic risk factors and XFS/XFG as such quantitative analyses integrate results from numerous studies on the topic of study, potentially drawing more objective and reliable conclusions. Here, we conducted a pooled analysis to evaluate the association between *LOXL1* gene polymorphisms and XFS/XFG susceptibility.

Three recent meta-analyses [34, 69, 70] investigated the association between the *LOXL1* gene polymorphisms and XFS/XFG risk. However, all of them covered papers published until to 2015, with the latest data unrepresented. Tang C et al. [69] and Chen H et al. [70] both

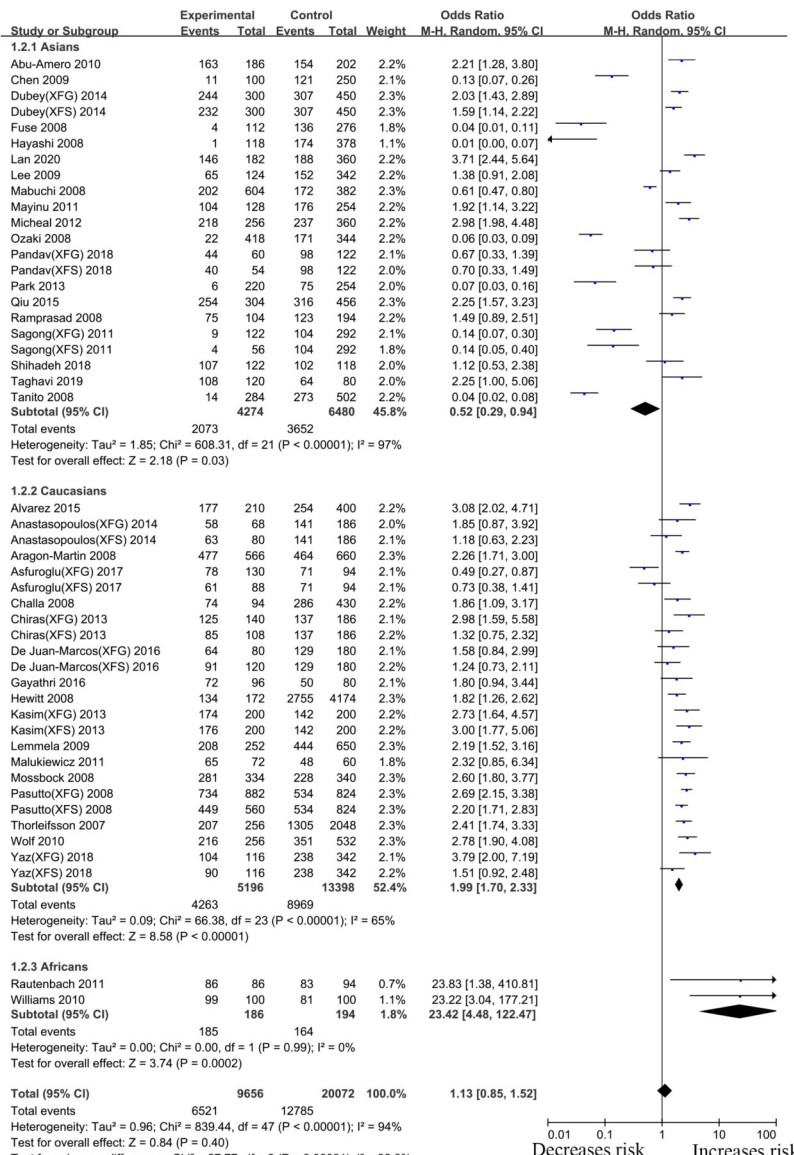

**Fig 3. Meta-analysis for the association between exfoliation syndrome/exfoliation glaucoma risks and LOXL1 gene polymorphism rs1048661 (G vs. T): Subgroup analysis by ethnicity (squares depict individual studies and diamonds depict summary effect size estimates (Odds Ratio, OR)).**

indicated that the allele G of rs1048661, the allele T of rs2165241 and the allele G of rs3825942 were associated with an increased risk for XFS/XFG among Caucasians, and that only the allele G of rs1048661 and the allele T of rs2165241 had a potential protective effect on XFS/XFG in Asians. Nevertheless, our study showed that there was no significant association between the LOXL1 gene rs2165241 polymorphism and XFS/XFG risk in Asians, and that rs3825942 ("G" allele) carriers are at higher risk of XFS/XFG relative to A allele carriers in Asians. On this point, our conclusion seems partially inconsistent with the previous two meta-analyses. More-over, their study did not involve XFS/XFG in Africans, which is important and worthy of attention. Wang L et al. [34] reported that rs1048661("G" alleles) had a weak association with XFG/XGS; rs3825942 ("G" alleles) had a strong association with XFS/XFG; and rs2165241

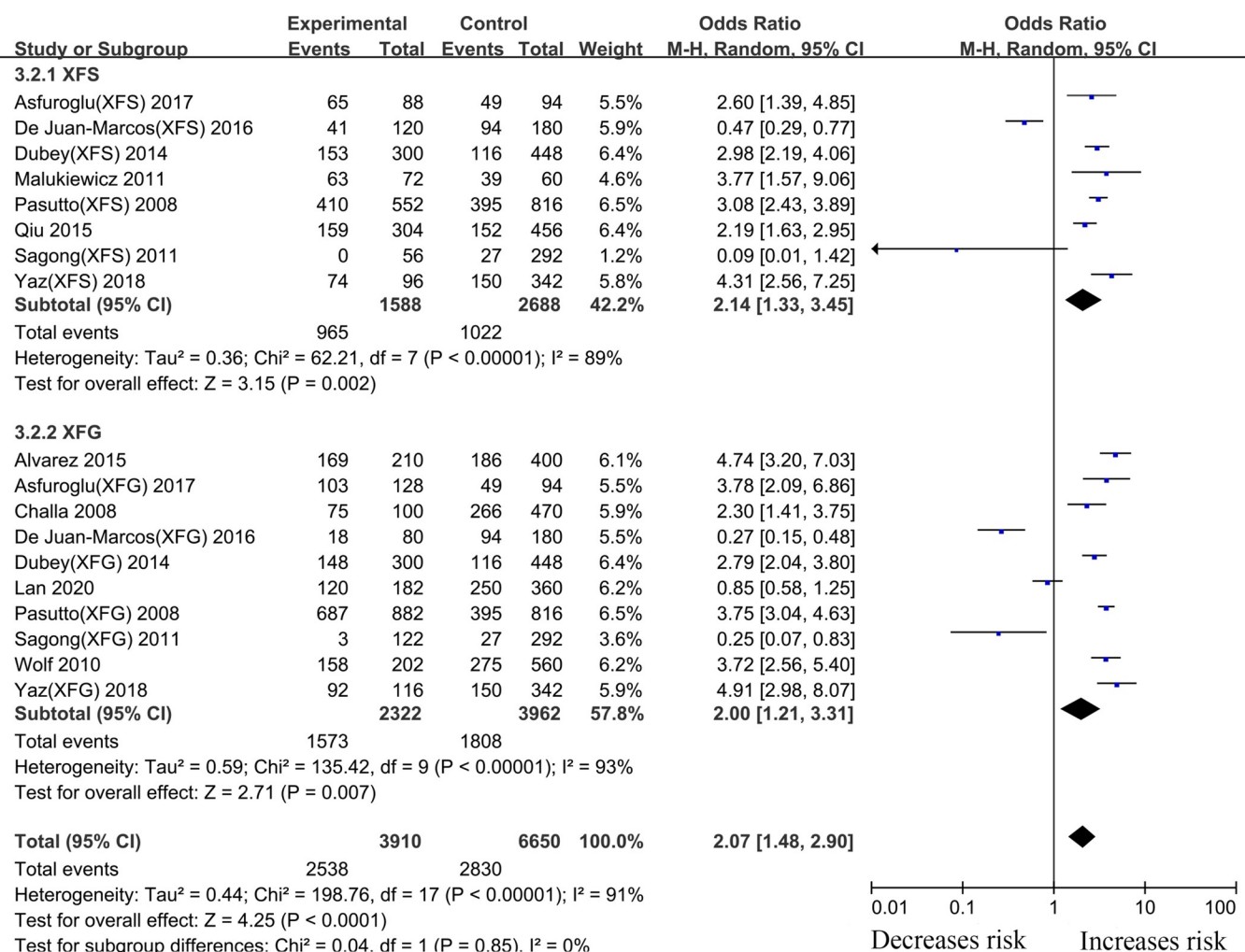

**Fig 4. Meta-analysis for the association between exfoliation syndrome/exfoliation glaucoma risks and LOXL1 gene polymorphism rs2165241 (T vs. C): Subgroup analysis by disease types (squares depict individual studies and diamonds depict summary effect size estimates (Odds Ratio, OR)).**

("T" alleles) had a significant risk with XFS/XFG in Caucasians. Our meta-analysis has corroborated their findings. However, three articles [10, 30, 31] included in Wang's meta-analysis [34], did not achieve HWE in the control group, while two articles [32, 33] examined the relationship between *LOXL1* gene polymorphisms and primary open-angle glaucoma. Here, we carried out an updated meta-analysis of the association between *LOXL1* gene polymorphisms and XFS/XFG susceptibility, involving 13984 participants. We identified three polymorphisms, rs1048661, rs3825942, and rs2165241, that met the inclusion criteria for meta-analysis. XFS/XFG analysis by ethnicity revealed a significantly high association between the G allele of rs1048661, the allele T of rs2165241 and the allele G of rs3825942, and XFS/XFG risk in Caucasians. We found that the G allele of rs1048661 may have potentially negative effects on XFS/XFG in Africans, and the G allele of rs3825942 may protect from XFS/XFG in Africans. In Asians, a significantly increased XFS/XFG risk was associated with the G allele of rs3825942. However, we also found that the G allele of rs1048661 was associated with reduced XFS/XFG risk in Asians. In Asians, there was no significant association between the T allele of rs2165241 and XFS/XFG risk. Additionally, there was a significant association between *LOXL1* gene

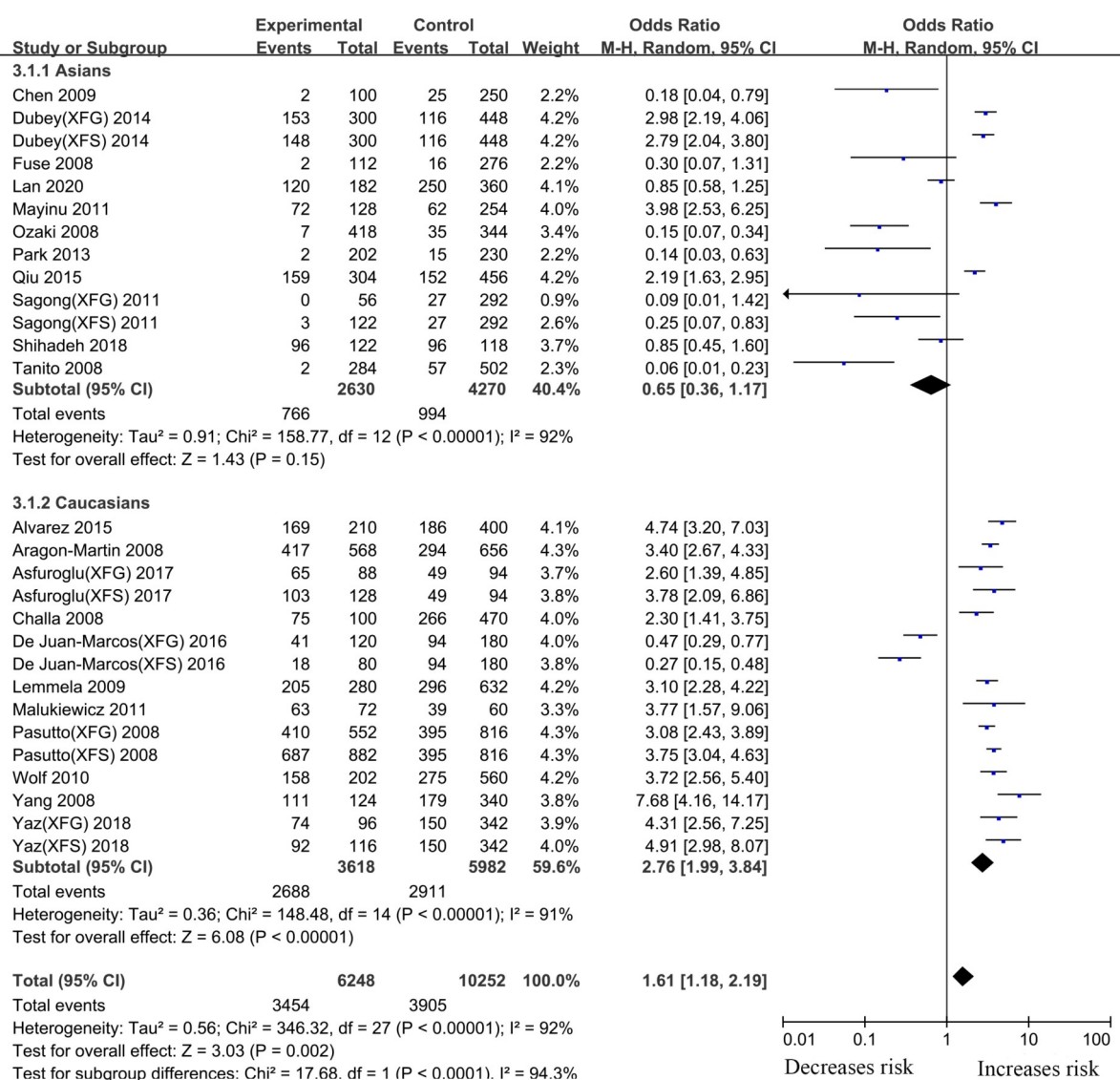

**Fig 5. Meta-analysis for the association between exfoliation syndrome/exfoliation glaucoma risks and LOXL1 gene polymorphism rs2165241 (G vs. A): Subgroup analysis by ethnicity (squares depict individual studies and diamonds depict summary effect size estimates (Odds Ratio, OR)).**

polymorphisms and susceptibility to various disease types. These results affirmed the association between *LOXL1* gene polymorphisms and XFS and XFG. Notably, we found a high frequency of risk alleles (rs1048661, rs2165241, and rs3825942) in non-XFG/XFS individuals, especially in Caucasians. Some studies have reported that these polymorphisms affect the proteolytic activity of LOXL1, and LOXL1 is an important matrix cross-linking enzyme that is required for elastic fiber formation and confer risk for the development of XFS/XFG [71]. However, the contribution of the risk alleles to XFS/XFG is complicated. Certain genetic variants of LOXL1, which has a prominent role in elastin fiber production, are not a single causative factor as many genetically affected individuals do not develop XFS or XFG [72]. It is likely that additional genetic or environmental factors modulate the penetrance of *LOXL1* susceptibility alleles [52]. This meta-analysis found that *LOXL1* gene polymorphisms may contribute to XFS/XFG susceptibility in different populations, and the differences in genetic susceptibility

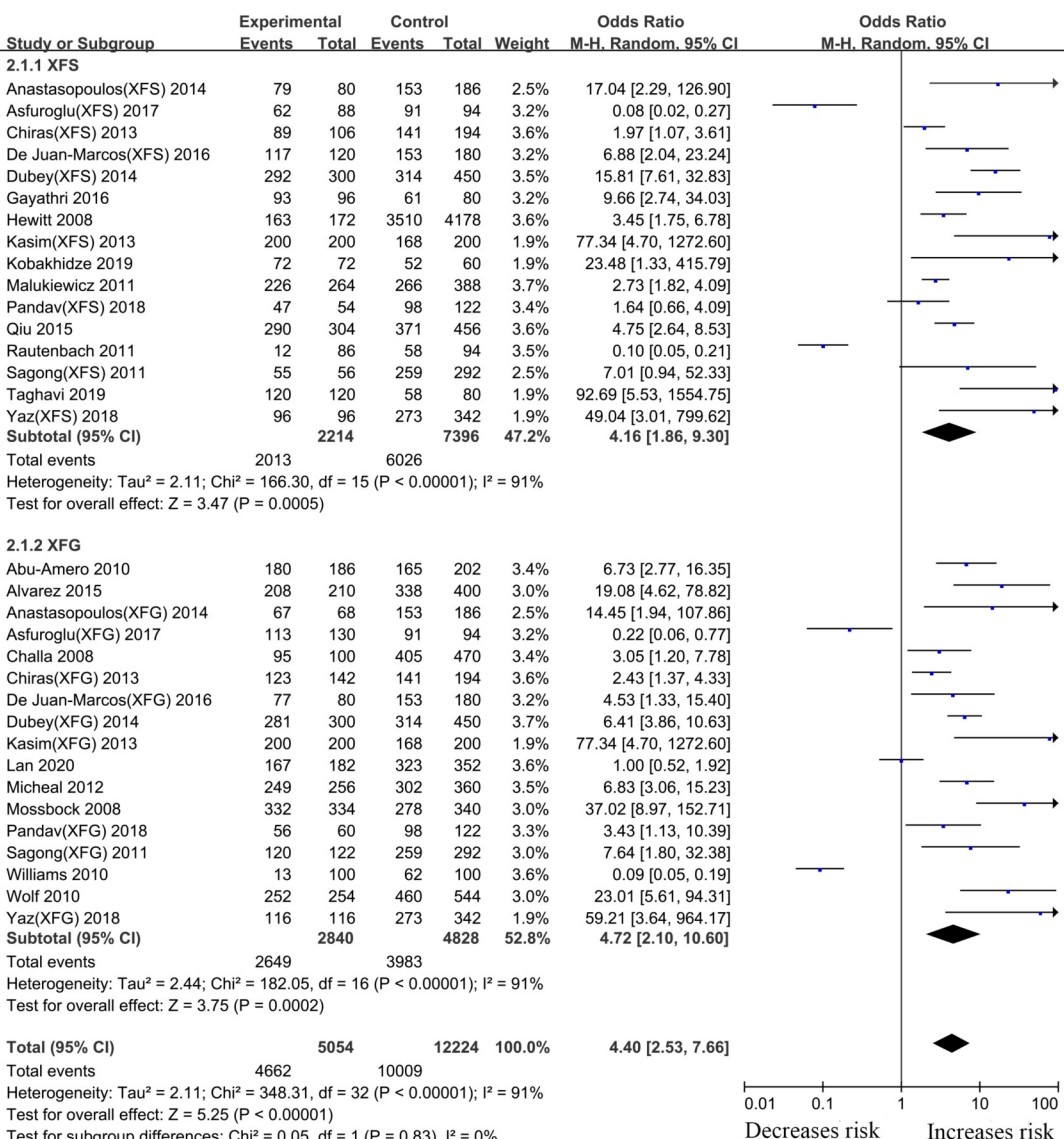

**Fig 6. Meta-analysis for the association between exfoliation syndrome/exfoliation glaucoma risks and LOXL1 gene polymorphism rs3825942 (G vs. A): Subgroup analysis by disease types (squares depict individual studies and diamonds depict summary effect size estimates (Odds Ratio, OR)).**

might be affected by ethnic factors, lifestyle factors and environmental exposures. Unfortunately, there are few studies concerning the association between *LOXL1* gene polymorphisms (rs3825942 and 1048661) and XFS/XFG in Africans, and no data were available for the SNP rs2165241 in Africans. This may lead to bias in the conclusion and generalization of the relationship between *LOXL1* gene polymorphisms and XFS/XFG in Africans. Thus, many such original studies are needed to confirm these findings as the currently included case-control studies are based on small sample sizes, especially for African populations.

The mechanisms by which *LOXL1* gene polymorphisms affect XFS/XFG susceptibility remain unclear. Multiple studies [63, 73] have shown that LOXL1 mediates the formation and maintenance of elastic tissues, as well as maintenance of extracellular matrix homeostasis, by regulating cross-linking reactions between collagen and elastin. LOXL1 has also been reported

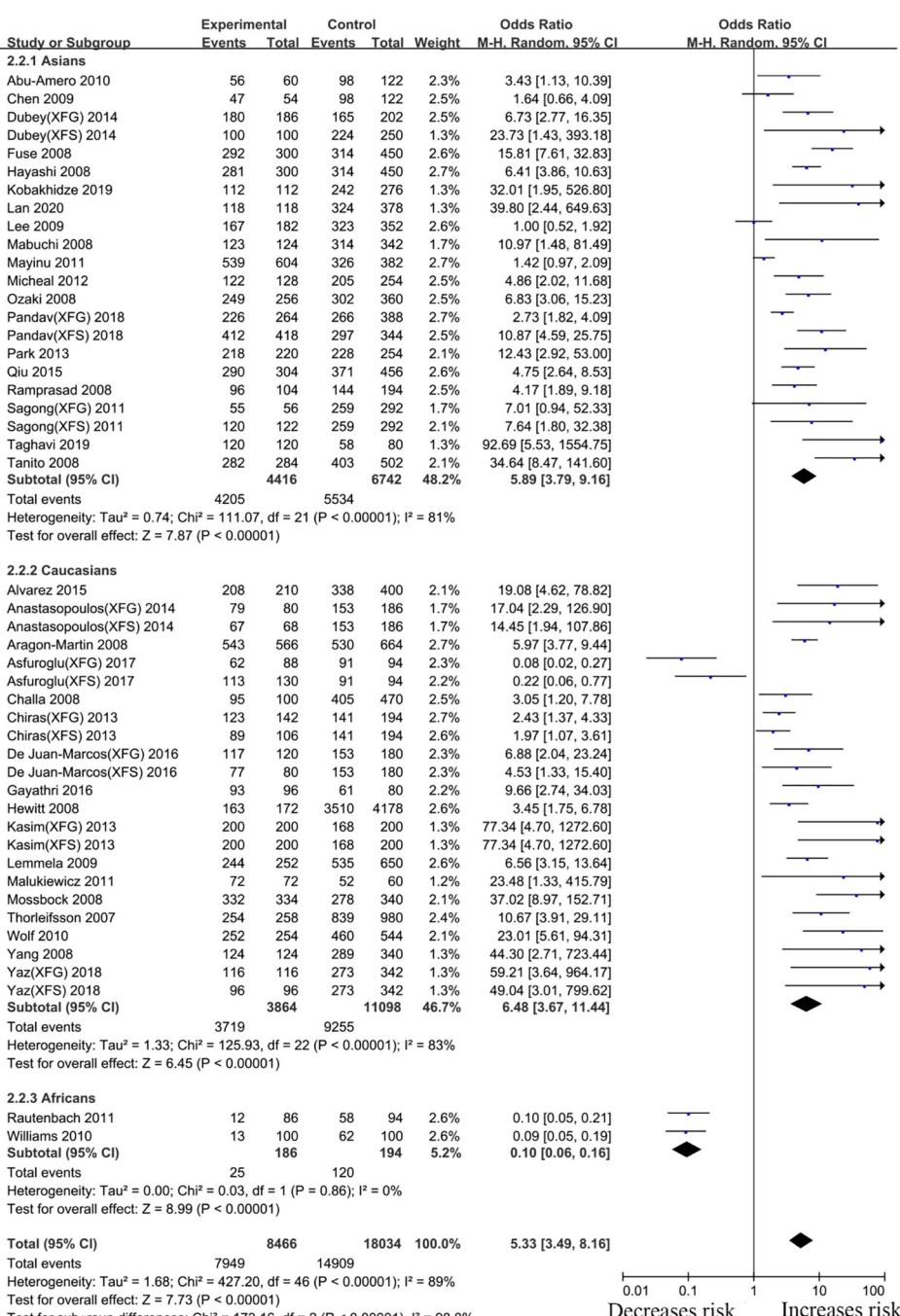

**Fig 7. Meta-analysis for the association between exfoliation syndrome/exfoliation glaucoma risks and LOXL1 gene polymorphism rs3825942 (G vs. A): Subgroup analysis by ethnicity (squares depict individual studies and diamonds depict summary effect size estimates (Odds Ratio, OR)).**

to be involved in elastin renewal and XFS/XFG development [71, 74]. Sharma et al. [59] reported that the coding variants rs1048861 and rs3825942 may alter protein function and binding, wherein molecular modeling displayed that positions 141(rs1048661) and 153 (rs3825942) of the LOXL1 protein are likely surface residues and hence possible recognition sites for protein-protein interactions. Alterations at these residues might change the capacity

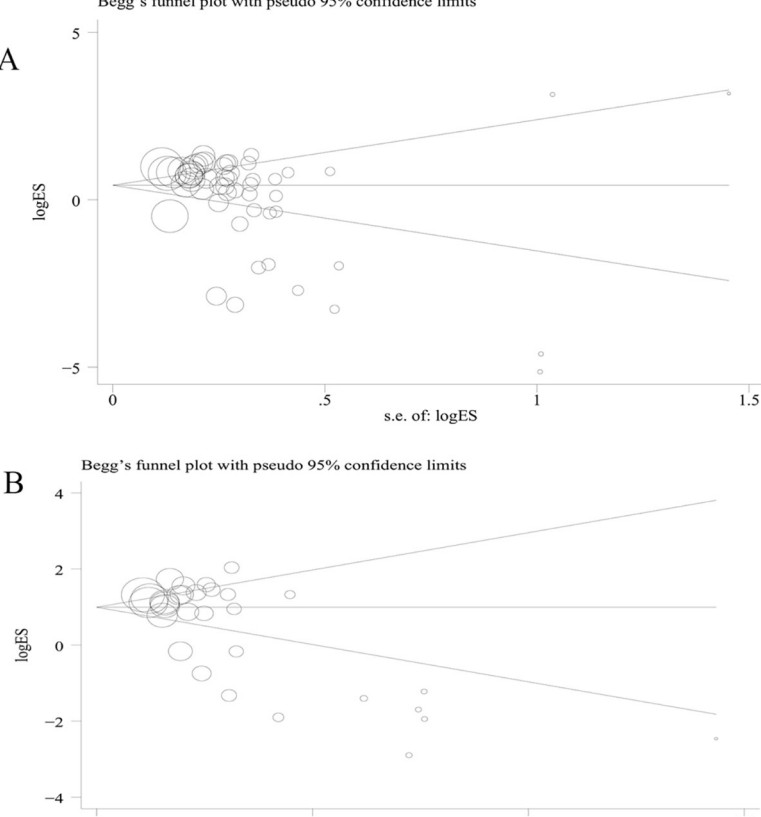

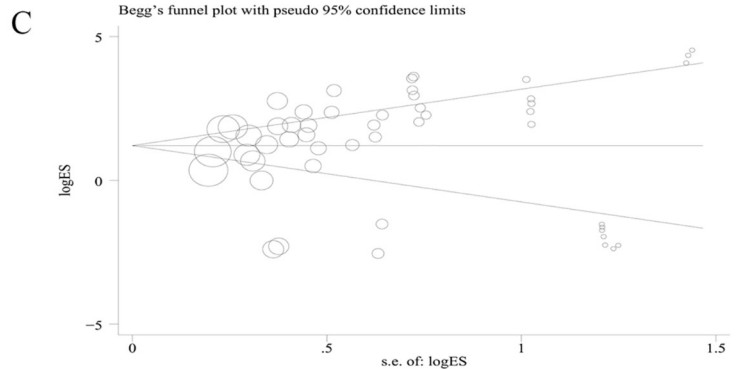

**Fig 8.** (A) Begg's funnel plot of publication bias for LOXL1 gene polymorphism rs1048661; (B) Begg's funnel plot of publication bias for LOXL1 gene polymorphism rs2165241; (C) Begg's funnel plot for of publication bias for LOXL1 gene polymorphism rs3825942.

of LOXL1 to bind other proteins related to its cleavage as well as processing. Nevertheless, the difference in processing of the LOXL1 protein variants detected in their research does not completely interpret susceptibility to XFS/XFG among carriers of these variants as each of the variants confer the XFS/XFG risk in various ethnicities. The detailed mechanism whereby *LOXL1* gene polymorphisms lead to the XFS/XFG, remains poorly understood. Therefore, further studies are required to elucidate the mechanism on how the *LOXL1* gene polymorphisms

impact the occurrence and development of XFS/XFG. Moreover, the distinct genetic background of Caucasians from Asians may modify LOXL1-mediated genetic susceptibility; hence, the effects of rs2165241 and rs1048661 are opposite in Asians and Caucasians. Genetic and/or environmental factors may modify the effects of gene polymorphisms in different ethnic groups.

High heterogeneity was found in our study. For exploring the underlying source of heterogeneity, a subgroup analysis and sensitivity analysis were performed. Unfortunately, although subgroup and sensitivity analyses were performed, obvious heterogeneity still existed in certain genetic models, and it is difficult to explain the heterogeneity completely. Thus, we speculated that living environment and other complications might lead to heterogeneity. Publication bias was assessed using Begg's funnel plot and Egger's test; no significant publication bias was found in this meta-analysis. Moreover, all genotype distributions of controls were in -absolute accordance with the HWE, indicating that our results are stable and reliable.

We acknowledge several limitations of this study. First, in subgroup analysis by ethnicity and disease type, some subgroups consisted of less than three case-control studies, which may be too small to detect associations. Second, data were not stratified by other factors, such as gender, age, gene-environment/gene-gene interactions, and lifestyle, because sufficient information could not be extracted from primary publications. Third, we mainly focused on *LOXL1* gene polymorphisms, and did not take into consideration potential linkage disequilibrium with other mutations in this gene, or gene-gene and gene-environment interactions. Moreover, language bias may have occurred as only articles published in Chinese or English were included in the study. However, we minimized the likelihood of bias using a rigorous protocol, study identification, data selection, and statistical analysis.

## Conclusion

In conclusion, our findings indicate that rs1048661, rs3825942, and rs2165241 *LOXL1* polymorphisms may contribute to XFS/XFG susceptibility, especially in Caucasians. Furthermore, well-designed studies with large sample sizes focusing on ethnicity or disease types are needed to confirm these findings.

## Supporting information

**S1 File. PRISMA 2009 checklist.**
(DOC)

## Acknowledgments

We sincerely appreciated to all authors contributed to this article and Editage for English language editing.

## Author Contributions

**Data curation:** Jie He.

**Formal analysis:** Xiaoyan Li.

**Methodology:** Xiaoyan Li.

**Project administration:** Jian Sun.

**Resources:** Jie He.

**Supervision:** Jian Sun.

**Validation:** Jie He.

**Writing – original draft:** Xiaoyan Li.

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
