## [Decision Letter · Decision Letter 0]

27 Jan 2021

PONE-D-20-37687

LOXL1 gene polymorphisms are associated with exfoliation syndrome/exfoliation glaucoma risk: an updated meta-analysis

PLOS ONE

Dear Dr. Jie,

Thank you for submitting your manuscript to PLOS ONE. After careful consideration, we feel that it has merit but does not fully meet PLOS ONE’s publication criteria as it currently stands. Therefore, we invite you to submit a revised version of the manuscript that addresses the points raised during the review process.

The reviewer has identified numerous inaccuracies in this manuscript, and much needs to be rewritten due to poor sentence structure and lack of clarity. If all of the numerous problems listed can be corrected, we will reconsider this manuscript for publication. It is also true that this topic has been reviewed extensively by other authors. It is important that you point out what this manuscript adds to the literature that has not already been presented by others.

We look forward to receiving your revised manuscript.

Kind regards,

Philip C. Trackman, Ph.D.

Academic Editor

PLOS ONE

Journal Requirements:

2. Please include your tables as part of your main manuscript and remove the individual files. Please note that supplementary tables should be uploaded as separate "supporting information" files.

"none"

"none"

6. Please ensure that you refer to Figure 1 in your text as, if accepted, production will need this reference to link the reader to the figure.

Reviewers' comments:

Reviewer's Responses to Questions

**Comments to the Author**

1. Is the manuscript technically sound, and do the data support the conclusions?

Reviewer #1: Yes

2. Has the statistical analysis been performed appropriately and rigorously? 

Reviewer #1: Yes

3. Have the authors made all data underlying the findings in their manuscript fully available?

Reviewer #1: Yes

4. Is the manuscript presented in an intelligible fashion and written in standard English?

Reviewer #1: Yes

5. Review Comments to the Author

Reviewer #1: The authors in this manuscript report a meta-analysis over the association of known risk alleles of the gene LOXL1 and XFS/XFG, including 36658 participants (cases and controls) of different ethnic groups. The research design is appropiate and the conclusions supported by the results. However, the authors should review the manuscript in depth. Mistakes and inaccuracies throughout the text must be addressed. References should be revised and some parts of the text should be rewritten or clarified. Moreover, extensive editing of English language is required.

Comments and Suggestions for Authors:

1. Human gene symbols must be italicized throughout the text. LOXL1 when you refer to the gene. LOXL1 for the protein.

2. Page 7, Abstract background, first sentence: “Lysyl oxidase-like 1 (LOXL1), a matrix cross-linking enzyme involved in elastic fiber formation, is a risk factor for exfoliation syndrome and exfoliation glaucoma” This sentence should be corrected.

Single nucleotide polymorphisms (SNPs) in the gene encoding the LOXL1 are the risk factors for exfoliation syndrome and exfoliation glaucoma”.

3. Page 8, Abstract results: “… was observed in the disease types-based subgroup” should be “were observed in the disease type-based subgroups”

4. Page 8, Abstract results: please clarify “…combined polymorphism…”

5. Page 8, Abstract results: “relative to” should be “related to”

6. Page 8, Introduction, paragraph 1: please clarify “occoecatio”. Is there another better known term with the same meaning?

7. Page 8, Introduction, paragraph 1: Reference 4 is before reference 3 in the text.

8. The article corresponding to reference number 3 in the list of references, has been retracted by the editor of International Journal of Ophthalmology. This reference should be removed from the article.

9. Page 9, paragraph 1: “intraocular pressure” should be “elevated intraocular pressure”

10. Page 9, paragraph 1: Add a comma “Relative to primary open-angle glaucoma, XFS-associated secondary open-angle glaucoma...”

Points 11-15 refer to the same paragraph of the introduction (Paragraph 3 of page 9 and 1 of page 10), which must be rewritten more neatly after an exhaustive revision. I think the authors try to show the discrepancies between some studies carried out in different populations of the same or different ethnic origin, but they fail to do it due to the mistakes and inaccuracies they commit.

11. Page 9, paragraph 3: reference 16 is wrong. The correct reference is: Thorleifsson G, Magnusson KP, Sulem P, Walters GB, Gudbjartsson DF, Stefansson H et al. Common sequence variants in the LOXL1 gene confer susceptibility to exfoliation glaucoma. Science. 2007;317(5843):1397-1400.

12. Page 9, paragraph 3: This statement is incorrect: “Dubey SK et al (17) reported that LOXL1 G (rs1048661) and T (rs3825942) alleles are XFS/XFG risk factors in Caucasians.” The work of Dubey et al was carried out in an Asian population no in Caucasians. Moreover, there is also some mistake regarding the risk alleles.

Dubey et al reported that the G allele of rs1048661, the G allele of rs3825942 and the T allele of rs2165241 were XFS/XFG risk factors in a South Indian population. Which is similar to that found in the original study conducted by Thorleifsson et al (2007) in Caucasians (Scandinavian population), as well as in most studies carried out in Caucasians, but different to that reported in most studies in Asians, where the alleles T and C of rs1048661 and rs2165241, respectively, are the risk alleles. References of these works should be added.

13. Pages 9-10, Paragraph 3 of page 9 and 1 of page 10: The sentence “De Juan-Marcos L et al (18) showed that LOXL1 rs2165241 polymorphism may be associated with XFS/XFG susceptibility” should be corrected.

De Juan-Marcos et al showed that the G allele of rs3825942 and the T allele of rs2165241 were XFS/XFG risk factors in a Spanish population. But no significant association between XFS/XFG and the SNP rs1048661 was observed. This last observation is the most relevant, since it shows a discrepancy with the majority of studies in Caucasian populations. References of these works should be added.

14. Page 10, paragraph 1: The statement “However, Tanito M et al (19) reported that LOXL1 G and T alleles are protective against XFS/XFG in Japanese populations. Similar observations were made by Chen L et al (20)” should be extended to more Asian studies, where G (rs1048661) and T (rs2165241) alleles are protective against XFS/XFG, and not only to the work of Chen L et al (20) in a Chinese population. References of these works should be added. See point 12.

15. Page 10, paragraph 1: Some reference should be made to the results found in the literature for the black race and discrepancies with other ethnicities.

16. Page 10, paragraph 2: The statement “Past genetic predisposition studies have been inconclusive” should be qualified.

Despite the existence of discrepancies between some studies related with the risk alleles of LOXL1 SNPs, it is widely accepted that LOXL1 gene is the most important genetic risk factor known so far for XFS/XFG.

17. Page 10, Material and methods: Please clarify the point “3) if they consisted of a complete number of genotypes among cases and controls”

18. Page 11, Statistical analyses: Please, replace “gene models” by “genetic models”

19. Page 11, Statistical analyses: Add software/s used to carry out the statistical analyses.

20. Page 12, Publication bias: References 22 and 23 are not the most suitable. I suggest the following references:

Peters JL, Sutton AJ, Jones DR, et al. Contour-enhanced meta-analysis funnel plots help distinguish publication bias from other causes of asymmetry. J Clin Epidemiol 2008;61:991–6.

Sterne J A, Gavaghan D, Egger M. Publication and related bias in meta-analysis: power of statistical tests and prevalence in the literature. J Clin Epidemiol 2000;53:1119–29.

21. Page 12, 3.1 Study characteristics: Please, correct Figure 1. According to the text, titles and abstracts where searched in step 1 and not in step 2 as showed in the Figure 1.

22. Page 12, 3.1 Study characteristics, line 6, “a previous meta-analysis” should be “in a previous meta-analysis”

23. Table 2: Please, specify what is N

24. I suppose that N is the number of studies, but in table 2, for rs1048661, N is 39 instead of 38 as showed in the text (page 13, line 1).

25. In Table 2 N is 17 for rs1048661, XFS, however, in Figure 2 only 16 studies are listed.

26. In Table 2 N is 21 for rs1048661 in Asians, however, in Figure 3 only 19 different studies are listed.

27. In Table 2 N is 18 for rs1048661 in Caucasians, however, in Figure 3 only 17 different studies are listed.

28. Please, clarify y/o correct the following discrepancies between the data of the tables 1 and 2:

- In table 2 the total number of cases (XFS/XFG) is 4900 for rs1048661, however, if the number of cases from each study listed in Table 1 are added, the total number of cases is 4828.

- In table 2 the total number of controls (XFS/XFG) is 10111 for rs1048661, however, if the number of controls from each study listed in Table 1 are added, the total number of cases is 10036.

- In table 2 the total number of cases (XFS/XFG) is 4305 for rs3825942, however, if the number of cases from each study listed in Table 1 are added, the total number of cases is 4233.

- In table 2 the total number of controls (XFS/XFG) is 9092 for rs3825942, however, if the number of controls from each study listed in Table 1 are added, the total number of cases is 9017.

29. Page 13, first paragraph: The authors claim that no associations were found between the SNP rs1048661 and XFS/XFG (total group). That is according to table 1 and Fig3 data but however, according to the data of figure 2, there is association. OR=1.73 �1.41, 2.12�.

Please, explain more in detail the two analysis realized for each SNP (by disease and by ethinicity) and why analysis considering ethnicity is chosen.

It should be clarify in the text that the data “G vs. T, OR:1.13,95%CI: 0.85-1.52, P:0.40” are shown in Fig3.

30. Page 13, second paragraph, “rs3825942” should be “rs2165241”.

31. Page 13, second paragraph, it should be clarify in the text that the data “T vs. C, OR: 1.61, 95%CI: 1.18-2.19, p:0.002” are shown in Fig5.

32. Page 13, second paragraph, “Fig4” should be “Fig4, Table 2”.

33. Page 14, first paragraph, “Fig5” should be “Fig5, Table 2”.

34. Page 14, second paragraph, it should be clarify in the text that the data “G vs. A, OR: 5.33, 95%CI: 3.49-8.16, p<0.001” are shown in Fig7.

35. Page 14, second paragraph, “Fig6” should be “Fig6, Table 2” and “Fig7” should be “Fig7, Table 2”

36. Page 14, second paragraph, the dot must be replaced by a comma. “…increased risks were identified among Caucasians (G vs. A, OR: 6.48, 95%CI: 3.67-11.44, P<0.001) and Asians (G vs. A, OR: 5.89, 95%CI: 3.79-9.16, p <0.001) (Fig 7), suggesting that…”

37. Figures 2-7 legends: The reader should be provided with more details about the figures. For instance, what the squares (or the diamonds) and their size indicate.

38. Figures 2-7 legends: Please, replace “LOXL1 gene polymorphism in rs…” by “LOXL1 gene polymorphism rs…”

39. Figure 8: Please, correct the leyend: Begg’s funnel plot for evaluation of publication bias in the selection of studies on the association between between exfoliation syndrome /exfoliation glaucoma risks and LOXL1 gene polymorphism, rs1048661(A), rs2165241(B) and rs3825942(C).

40. Page 15, Discussion: The statement “Various population studies have provided inconclusive evidence that LOXL1 single nucleotide polymorphisms are strongly

associated with XFS/XFG” should be qualified.

As mentioned in point 18, despite the existence of discrepancies between some studies related with the risk alleles of LOXL1 SNPs, it is widely accepted that LOXL1 gene is the most important genetic risk factor known so far for XFS/XFG.

Moreover, references 6, 70, 71 are not suitable.

41. Page 15, Discussion: Please, remove the reference 3 which has been retracted by the editor of International Journal of Ophthalmology.

42. Page 16, line 4: The number of participants was 36658 (cases + controls)

43. Page 16: Risk alleles should be specified in the sentence “XFS/XFG analysis by ethnicity revealed significantly high association between the 3 LOXL1 polymorphisms and XFS/XFG risk in Caucasians”.

“XFS/XFG analysis by ethnicity revealed significantly high association between the allele G of rs1048661, the allele T of rs2165241 and the allele G of rs3825942 and XFS/XFG risk in Caucasians”

44. The same should be done in the rest of the paragraph with all SNPs, since alleles that confer risk vary between ethnic groups. The risk alleles should be specified. For instance: “…the variant G of rs1048661 polymorphism may have potentially negative effects on…”

Moreover, some mistakes should be corrected: For instance: “In Asians, significantly increased XFS/XFG risk was associated with the two LOXL1 polymorphisms (rs3825942)” should be “In Asians, significantly increased XFS/XFG risk was associated with the allele G of rs3825942”.

Please, revise, correct and rewrite the paragraph in a more orderly way.

45. The lack of studies in Africans, specially about the SNP rs2165241 should be mentioned in the text.

46. In my opinion, mention should be made of the high frequencies of risk alleles in the controls and discuss their implications.

47. Page 16, Discussion: Differences and similarities with the results of other studies or with previous meta-analysis must be described and discussed in the manuscript. Does this work make any new contribution to the field or confirm the results of previous studies?

48. Page 16: Clarify the sentence: “…the differences in genetic susceptibility might be affected by ethnic factors and periods of glaucoma”.

Ethnic factors can affect genetic susceptibility, the genetic background is different between ethnicities. However, the stage of glaucoma can not affect genetic susceptibility. The genotype of an individual do not change, is the same throughout his life regardless of the stage of a disease

49. Page 17: Please, provide details of the studies that could support the hypothesis: “We hypothesized that the LOXL1 gene polymorphisms may upregulate LOXL1 production and that the high serum concentration of LOXL1 may contribute to XFS/XFG susceptibility”

50. Page 17: Please, correct the sentence“… hence the observed opposite rs1048661 risks in Asian vs Caucasians subjects”.

The effect of rs2165241 is also opposite in Asian and Caucasian

51. Page 17: Please, rewrite the sentence: “Heterogeneity involved in studies is generally in meta-analysis for the quality of the included research, population characteristics, experimental methods and other reasons”.

52. Page 17: Please, correct or clarify the sentence: “Thus, we speculated that there are several other factors leading to heterogeneity: Blood sample sampling and storage methods are different; the measurement method and experimental conditions are different; varying levels of illness severity”.

In my opinion the factors leading to heterogeneity listed by the authors are irrelevant to a genetic study. The genotype of an individual is not affected by factors such as blood sample sampling or storage method. The mentioned factors could be relevant for other types of studies, for instance when quantification of a protein in blood is required.

6. PLOS authors have the option to publish the peer review history of their article (what does this mean?). If published, this will include your full peer review and any attached files.

Reviewer #1: No

---

## [Author Response · Author response to Decision Letter 0]

24 Feb 2021

Reply: The authors would like to thank the Reviewers and the Editor for their time in evaluating our study and for making useful suggestions. We believe that we have addressed all of the comments below, but we are open to any additional ones. 

Reply to Editors： 

Reply: We have revised the manuscript to meet PLOS ONE's style requirements.

2. Please include your tables as part of your main manuscript and remove the individual files. Please note that supplementary tables should be uploaded as separate "supporting information" files.

Reply: We have included the tables in my main manuscript and removed the individual files. 

3. Thank you for stating the following financial disclosure: "none"

1. Please clarify the sources of funding (financial or material support) for your study. List the grants or organizations that supported your study, including funding received from your institution.

2. State what role the funders took in the study. If the funders had no role in your study, please state: “The funders had no role in study design, data collection and analysis, decision to publish, or preparation of the manuscript.”

3. If any authors received a salary from any of your funders, please state which authors and which funders.

4. If you did not receive any funding for this study, please state: “The authors received no specific funding for this work.”

Reply: We have added the state in manuscript in Page 21 in manuscript.

"none"

Reply: We have added the state in manuscript in Page 21 in manuscript and cover letter.

Reply: All relevant data are within the paper and its Supporting information files.

7. Please include captions for your Supporting Information files at the end of your manuscript, and update any in-text citations to match accordingly. 

 Reply: All relevant Supporting Information files are within the paper and updated the citations one by one.

Reply to Reviewers： 

Reply: We thank the reviewers’ comments and excellent suggestions. We have revised these mistakes and inaccuracies in the text as you pointed out, rewritten some references and revised some parts of the text. In addition, we have consulted a professional company Editage for proofreading to polished the language in the manuscript. Reply to comments and suggestions of reviewers as follows:

1. Human gene symbols must be italicized throughout the text. LOXL1 when you refer to the gene. LOXL1 for the protein.

Reply: We have italicized throughout the text of the LOXL1 gene.

2. Page 7, Abstract background, first sentence: “Lysyl oxidase-like 1 (LOXL1), a matrix cross-linking enzyme involved in elastic fiber formation, is a risk factor for exfoliation syndrome and exfoliation glaucoma” This sentence should be corrected.

Single nucleotide polymorphisms (SNPs) in the gene encoding the LOXL1 are the risk factors for exfoliation syndrome and exfoliation glaucoma”.

Reply:We have revised the sentence in Page 1 in manuscript.

3. Page 8, Abstract results: “… was observed in the disease types-based subgroup” should be “were observed in the disease type-based subgroups”

Reply:We have revised the sentence in Page 2 in manuscript.

4. Page 8, Abstract results: please clarify “…combined polymorphism…”

Reply: We have replaced the words to a more proper sentence in Page 2 in manuscript.

5. Page 8, Abstract results: “relative to” should be “related to”

Reply: We have revised the words in Page 2 in manuscript.

6. Page 8, Introduction, paragraph 1: please clarify “occoecatio”. Is there another better known term with the same meaning?

Reply: We have replace the word to “blindness” in Page 2 in manuscript.

7. Page 8, Introduction, paragraph 1: Reference 4 is before reference 3 in the text.

Reply: We have removed the reference 4 to ahead of the former 3 in Page 2 in manuscript.

8. The article corresponding to reference number 3 in the list of references, has been retracted by the editor of International Journal of Ophthalmology. This reference should be removed from the article.

Reply: We have removed the reference in the article in Page 2 in manuscript.

9. Page 9, paragraph 1: “intraocular pressure” should be “elevated intraocular pressure”

Reply: We have revised the sentence in Page 3 in manuscript.

10. Page 9, paragraph 1: Add a comma “Relative to primary open-angle glaucoma, XFS-associated secondary open-angle glaucoma...”

Reply: We have added a comma in the sentence in Page 3 in manuscript.

11. Page 9, paragraph 3: reference 16 is wrong. The correct reference is: Thorleifsson G, Magnusson KP, Sulem P, Walters GB, Gudbjartsson DF, Stefansson H et al. Common sequence variants in the LOXL1 gene confer susceptibility to exfoliation glaucoma. Science. 2007;317(5843):1397-1400.

Reply: We have revised the reference, and revised the number of the following reference.

12. Page 9, paragraph 3: This statement is incorrect: “Dubey SK et al (17) reported that LOXL1 G (rs1048661) and T (rs3825942) alleles are XFS/XFG risk factors in Caucasians.” The work of Dubey et al was carried out in an Asian population no in Caucasians. Moreover, there is also some mistake regarding the risk alleles.

Dubey et al reported that the G allele of rs1048661, the G allele of rs3825942 and the T allele of rs2165241 were XFS/XFG risk factors in a South Indian population. Which is similar to that found in the original study conducted by Thorleifsson et al (2007) in Caucasians (Scandinavian population), as well as in most studies carried out in Caucasians, but different to that reported in most studies in Asians, where the alleles T and C of rs1048661 and rs2165241, respectively, are the risk alleles. References of these works should be added.

Reply: This statement is incorrect: “Dubey SK et al. reported that LOXL1 G (rs1048661) and T (rs3825942) alleles are XFS/XFG risk factors in Caucasians.” Has been re-written to “Dubey SK et al. reported that LOXL1 G (rs1048661), G (rs3825942) and T (rs2165241) alleles are XFS/XFG risk factors in Asians. Dubey SK et al' results is similar to that found in the original study conducted by Thorleifsson et al (2007) in Caucasians (Scandinavian population), as well as in most studies carried out in Caucasians” in Page 4 in manuscript.

13. Pages 9-10, Paragraph 3 of page 9 and 1 of page 10: The sentence “De Juan-Marcos L et al (18) showed that LOXL1 rs2165241 polymorphism may be associated with XFS/XFG susceptibility” should be corrected.

De Juan-Marcos et al showed that the G allele of rs3825942 and the T allele of rs2165241 were XFS/XFG risk factors in a Spanish population. But no significant association between XFS/XFG and the SNP rs1048661 was observed. This last observation is the most relevant, since it shows a discrepancy with the majority of studies in Caucasian populations. References of these works should be added.

Reply: It has been corrected to“Moreover, De Juan-Marcos et al showed that the G allele of rs3825942 and the T allele of rs2165241 were XFS/XFG risk factors in a Spanish population. But no significant association between XFS/XFG and the SNP rs1048661 was observed” in Page 4 in manuscript.

14. Page 10, paragraph 1: The statement “However, Tanito M et al (19) reported that LOXL1 G and T alleles are protective against XFS/XFG in Japanese populations. Similar observations were made by Chen L et al (20)” should be extended to more Asian studies, where G (rs1048661) and T (rs2165241) alleles are protective against XFS/XFG, and not only to the work of Chen L et al (20) in a Chinese population. References of these works should be added. See point 12.

Reply: It has been corrected to “However, different to that reported in most studies in Asians, where the alleles T and C of rs1048661 and rs2165241, respectively, are the risk alleles. Tanito M et al., Ozaki, Fuse, Hayashi, reported that the alleles T of rs1048661 as well as the alleles C of rs2165241 is associated with increased risks of XFS/XFG in Japanese . Park DY et al., Sagong, found a similar phenomenon in Koreans. Similar observations were made by Chen L et al in Chinese” in Page 4 in manuscript.

15. Page 10, paragraph 1: Some reference should be made to the results found in the literature for the black race and discrepancies with other ethnicities.

Reply: It has been corrected to “In addition, Rautenbach et al. and Williams et al. both indicated that the G allele of rs3825942 was protective in Black South Africans, and the G allele of rs1048661 was at risk allele for XFS/XFG” in Page 4 in manuscript.

16. Page 10, paragraph 2: The statement “Past genetic predisposition studies have been inconclusive” should be qualified.

Despite the existence of discrepancies between some studies related with the risk alleles of LOXL1 SNPs, it is widely accepted that LOXL1 gene is the most important genetic risk factor known so far for XFS/XFG.

Reply: We have revised the sentence of the state in Page 4 in manuscript.

17. Page 10, Material and methods: Please clarify the point “3) if they consisted of a complete number of genotypes among cases and controls”

Reply: We have revised the sentence to “they had complete genotype frequency data” in Page 5 in manuscript.

18. Page 11, Statistical analyses: Please, replace “gene models” by “genetic models”

Reply: We have revised the sentence in Page 6 in manuscript.

19. Page 11, Statistical analyses: Add software/s used to carry out the statistical analyses.

Reply: We have added the software about statistical analysis in Page 6 in manuscript..

20. Page 12, Publication bias: References 22 and 23 are not the most suitable. I suggest the following references:

Peters JL, Sutton AJ, Jones DR, et al. Contour-enhanced meta-analysis funnel plots help distinguish publication bias from other causes of asymmetry. J Clin Epidemiol 2008;61:991–6.

Sterne J A, Gavaghan D, Egger M. Publication and related bias in meta-analysis: power of statistical tests and prevalence in the literature. J Clin Epidemiol 2000;53:1119–29.

Reply: We have revised the two references as you recommended.

21. Page 12, 3.1 Study characteristics: Please, correct Figure 1. According to the text, titles and abstracts where searched in step 1 and not in step 2 as showed in the Figure 1.

Reply: We have revised the Figure 1 according to the text in Page 6 in manuscript.

22. Page 12, 3.1 Study characteristics, line 6, “a previous meta-analysis” should be “in a previous meta-analysis”

Reply: We have revised the sentence in Page 7 in manuscript.

23. Table 2: Please, specify what is N

Reply: We have replaced the N to Studies in Page 7 in manuscript.

24. I suppose that N is the number of studies, but in table 2, for rs1048661, N is 39 instead of 38 as showed in the text (page 13, line 1).

Reply: We have checked the numbers that the studies of rs1048661 is 38 indeed in Page 12 in manuscript.

25. In Table 2 N is 17 for rs1048661, XFS, however, in Figure 2 only 16 studies are listed.

Reply: We have revised the N in Table 2 in Page 7 in manuscript.

26. In Table 2 N is 21 for rs1048661 in Asians, however, in Figure 3 only 19 different studies are listed.

Reply: We have revised the N in Table 2 in Page 7 in manuscript.

27. In Table 2 N is 18 for rs1048661 in Caucasians, however, in Figure 3 only 17 different studies are listed.

Reply: We have revised the N in Table 2 in Page 7 in manuscript.

28. Please, clarify y/o correct the following discrepancies between the data of the tables 1 and 2:

- In table 2 the total number of cases (XFS/XFG) is 4900 for rs1048661, however, if the number of cases from each study listed in Table 1 are added, the total number of cases is 4828.

- In table 2 the total number of controls (XFS/XFG) is 10111 for rs1048661, however, if the number of controls from each study listed in Table 1 are added, the total number of cases is 10036.

- In table 2 the total number of cases (XFS/XFG) is 4305 for rs3825942, however, if the number of cases from each study listed in Table 1 are added, the total number of cases is 4233.

- In table 2 the total number of controls (XFS/XFG) is 9092 for rs3825942, however, if the number of controls from each study listed in Table 1 are added, the total number of cases is 9017.

Reply: We have revised the N in Table 2 in Page 7 in manuscript.

29. Page 13, first paragraph: The authors claim that no associations were found between the SNP rs1048661 and XFS/XFG (total group). That is according to table 1 and Fig3 data but however, according to the data of figure 2, there is association. OR=1.73-1.41, 2.12.

Please, explain more in detail the two analysis realized for each SNP (by disease and by ethinicity) and why analysis considering ethnicity is chosen.

It should be clarify in the text that the data “G vs. T, OR:1.13,95%CI: 0.85-1.52, P:0.40” are shown in Fig3.

Reply: We have clarified the reason of the discrepancy is that “Some studies included patients with XFS and XFG, but did not distinguish which were XFS patients, and which were XFG patients. Thus, in the subgroup analysis based on the type of disease, we only extracted data from studies in which disease types (XFS or XFG) are clearly illustrated. While, in the subgroup analysis based on ethnicity, we combine all types of studies (XFS, XFG, XFS/XFG) to conduct a meta-analysis. For the reason that analysis realized for SNP by ethnicity is more comprehensive, we choose its merger result as the overall result” in Page 11 in manuscript. 

30. Page 13, second paragraph, “rs3825942” should be “rs2165241”.

Reply: We have revised the mistake in Page 14 in manuscript.

31. Page 13, second paragraph, it should be clarify in the text that the data “T vs. C, OR: 1.61, 95%CI: 1.18-2.19, p:0.002” are shown in Fig5.

Reply: We have clarified the question in Page 14 in manuscript.

32. Page 13, second paragraph, “Fig4” should be “Fig4, Table 2”.

Reply: We have added the Table2 in Page 14 in manuscript.

33. Page 14, first paragraph, “Fig5” should be “Fig5, Table 2”.

Reply: We have added the Table2 in Page 14 in manuscript.

34. Page 14, second paragraph, it should be clarify in the text that the data “G vs. A, OR: 5.33, 95%CI: 3.49-8.16, p<0.001” are shown in Fig7.

Reply: We have clarified the question in Page 14 in manuscript.

35. Page 14, second paragraph, “Fig6” should be “Fig6, Table 2” and “Fig7” should be “Fig7, Table 2”

Reply: We have added the Table2 in Page 14 in manuscript.

36. Page 14, second paragraph, the dot must be replaced by a comma. “…increased risks were identified among Caucasians (G vs. A, OR: 6.48, 95%CI: 3.67-11.44, P<0.001) and Asians (G vs. A, OR: 5.89, 95%CI: 3.79-9.16, p <0.001) (Fig 7), suggesting that…”

Reply: We have revised the dot to comma in the sentence in Page 15 in manuscript.

37. Figures 2-7 legends: The reader should be provided with more details about the figures. For instance, what the squares (or the diamonds) and their size indicate.

Reply: Squares depict individual studies and diamonds depict summary effect size estimates (Odds Ratio, OR).

38. Figures 2-7 legends: Please, replace “LOXL1 gene polymorphism in rs…” by “LOXL1 gene polymorphism rs…”

Reply: “LOXL1 gene polymorphism in rs…”is replaced by “LOXL1 gene polymorphism rs…”

39. Figure 8: Please, correct the leyend: Begg’s funnel plot for evaluation of publication bias in the selection of studies on the association between between exfoliation syndrome /exfoliation glaucoma risks and LOXL1 gene polymorphism, rs1048661(A), rs2165241(B) and rs3825942(C).

Reply: It could be corrected “Figure 8: (A)Begg’s funnel plot of publication bias for LOXL1 gene polymorphism rs1048661; (B) Begg’s funnel plot of publication bias for LOXL1 gene polymorphism rs2165241; (C) Begg’s funnel plot for of publication bias for LOXL1 gene polymorphism rs3825942”.

40. Page 15, Discussion: The statement “Various population studies have provided inconclusive evidence that LOXL1 single nucleotide polymorphisms are strongly

associated with XFS/XFG” should be qualified.

As mentioned in point 18, despite the existence of discrepancies between some studies related with the risk alleles of LOXL1 SNPs, it is widely accepted that LOXL1 gene is the most important genetic risk factor known so far for XFS/XFG.

Moreover, references 6, 70, 71 are not suitable.

Reply: We have deleted the statement “Various population studies have provided inconclusive evidence that LOXL1 single nucleotide polymorphisms are strongly

associated with XFS/XFG” in Page 16 in manuscript and revised the references.

41. Page 15, Discussion: Please, remove the reference 3 which has been retracted by the editor of International Journal of Ophthalmology.

Reply: We have removed the reference 3 which has been retracted by International Journal of Ophthalmology in Page 2 in manuscript.

42. Page 16, line 4: The number of participants was 36658 (cases + controls)

Reply: There is a mistake about the number, and we have revised it in Page 18 in manuscript..

43. Page 16: Risk alleles should be specified in the sentence “XFS/XFG analysis by ethnicity revealed significantly high association between the 3 LOXL1 polymorphisms and XFS/XFG risk in Caucasians”.

“XFS/XFG analysis by ethnicity revealed significantly high association between the allele G of rs1048661, the allele T of rs2165241 and the allele G of rs3825942 and XFS/XFG risk in Caucasians”

Reply: The sentence has been revised in Page 18 in manuscript. 

44. The same should be done in the rest of the paragraph with all SNPs, since alleles that confer risk vary between ethnic groups. The risk alleles should be specified. For instance: “…the variant G of rs1048661 polymorphism may have potentially negative effects on…”

Moreover, some mistakes should be corrected: For instance: “In Asians, significantly increased XFS/XFG risk was associated with the two LOXL1 polymorphisms (rs3825942)” should be “In Asians, significantly increased XFS/XFG risk was associated with the allele G of rs3825942”.

Please, revise, correct and rewrite the paragraph in a more orderly way.

Reply: We have revised the sentence in Page 18 in manuscript.

45. The lack of studies in Africans, specially about the SNP rs2165241 should be mentioned in the text.

Reply: I have mentioned the lack of studies in Africans, specially about the SNP rs2165241 in Page 19 in manuscript. 

46. In my opinion, mention should be made of the high frequencies of risk alleles in the controls and discuss their implications.

Reply: I have mentioned the high frequencies of risk alleles in the controls and discuss their implications in Page 19 in manuscript. 

47. Page 16, Discussion: Differences and similarities with the results of other studies or with previous meta-analysis must be described and discussed in the manuscript. Does this work make any new contribution to the field or confirm the results of previous studies?

Reply: I have discussed differences and similarities with the results of other studies or with previous meta-analysis in Page 17 in manuscript.

48. Page 16: Clarify the sentence: “…the differences in genetic susceptibility might be affected by ethnic factors and periods of glaucoma”.

Ethnic factors can affect genetic susceptibility, the genetic background is different between ethnicities. However, the stage of glaucoma can not affect genetic susceptibility. The genotype of an individual do not change, is the same throughout his life regardless of the stage of a disease

Reply: “…the differences in genetic susceptibility might be affected by ethnic factors and periods of glaucoma” should be revised to “…the differences in genetic susceptibility might be affected by ethnic factors” in Page 19 in manuscript. 

49. Page 17: Please, provide details of the studies that could support the hypothesis: “We hypothesized that the LOXL1 gene polymorphisms may upregulate LOXL1 production and that the high serum concentration of LOXL1 may contribute to XFS/XFG susceptibility”.

Reply: We added two references, [76] and [77].

50. Page 17: Please, correct the sentence“… hence the observed opposite rs1048661 risks in Asian vs Caucasians subjects”.

The effect of rs2165241 is also opposite in Asian and Caucasian

Reply: it is corrected to “the effect of rs2165241 and rs1048661 are opposite in Asian and Caucasian” in Page 19 in manuscript. 

51. Page 17: Please, rewrite the sentence: “Heterogeneity involved in studies is generally in meta-analysis for the quality of the included research, population characteristics, experimental methods and other reasons”.

Reply: Delete this sentence: “Heterogeneity involved in studies is generally in meta-analysis for the quality of the included research, population characteristics, experimental methods and other reasons” . 

52. Page 17: Please, correct or clarify the sentence: “Thus, we speculated that there are several other factors leading to heterogeneity: Blood sample sampling and storage methods are different; the measurement method and experimental conditions are different; varying levels of illness severity”.

In my opinion the factors leading to heterogeneity listed by the authors are irrelevant to a genetic study. The genotype of an individual is not affected by factors such as blood sample sampling or storage method. The mentioned factors could be relevant for other types of studies, for instance when quantification of a protein in blood is required. 

Reply: Thus, we speculated that living environment and other complications might lead to heterogeneity in Page 20 in manuscript.

---

## [Decision Letter · Decision Letter 1]

11 Mar 2021

PONE-D-20-37687R1

LOXL1 gene polymorphisms are associated with exfoliation syndrome/exfoliation glaucoma risk: an updated meta-analysis

PLOS ONE

Dear Dr. Jie,

Thank you for submitting your manuscript to PLOS ONE. After careful consideration, we feel that it has merit but does not fully meet PLOS ONE’s publication criteria as it currently stands. Therefore, we invite you to submit a revised version of the manuscript that addresses the points raised during the review process.

The revised manuscript is improved, but the reviewer has identified remaining significant errors that need to be corrected. If these errors can be addressed we would review a revised manuscript submitted according to journal instructions.  

We look forward to receiving your revised manuscript.

Kind regards,

Philip C. Trackman, Ph.D.

Academic Editor

PLOS ONE

Journal Requirements:

Additional Editor Comments (if provided):

The reviewer has noted an improvement in the manuscript, but several significant errors remain that need to be corrected. If you can adress these issues we will reconsider a revised manuscript with changes made tot he manuscript explained according to journal requirements.

Reviewers' comments:

Reviewer's Responses to Questions

**Comments to the Author**

1. If the authors have adequately addressed your comments raised in a previous round of review and you feel that this manuscript is now acceptable for publication, you may indicate that here to bypass the “Comments to the Author” section, enter your conflict of interest statement in the “Confidential to Editor” section, and submit your "Accept" recommendation.

Reviewer #1: (No Response)

2. Is the manuscript technically sound, and do the data support the conclusions?

Reviewer #1: Yes

3. Has the statistical analysis been performed appropriately and rigorously? 

Reviewer #1: Yes

4. Have the authors made all data underlying the findings in their manuscript fully available?

Reviewer #1: Yes

5. Is the manuscript presented in an intelligible fashion and written in standard English?

Reviewer #1: Yes

6. Review Comments to the Author

Reviewer #1: The Authors made several important changes that increased the clarity of the manuscript. However, the authors should address following issues.

Comments and Suggestions for Authors:

1. Pag 1, Abstract Background, line 1: LOXL1 (do not use italics when referring to protein)

2. Pag 1, Abstract Results: The number of controls in the sentence “In total, 5022 cases and 10400 controls were included in this meta-analysis” should be corrected.

Table 1 show the number of XFS and XFG cases and controls in each study. Some studies report the results for XFS and XFG separately, in these studies the control group is the same for both diseases, but it was counted in duplicate. For example, in the case of the study of Panday et al., there is only a group control with 61 individuals.

For the same reason, the total number of participants must be corrected in Discussion, pag 19, first line.

3. Pag 2, Abstract results: Delete the word “only” in the sentence: “In addition, only rs1048661 and rs3825942 correlated with XFS/XFG susceptibility in Africans.

The SNP rs2165241 was not analized in Africans. Authors do not know if this SNP is associated with XFS/XFG.

4. Pag 3, Introduction: in the sentence “…as well as in most studies carried out in Caucasians [16,17]” delete reference 17, and add references of other studies in Caucasians

5. Pag 3-4, Introduction: Delete the word “where” in the sentence: “However, in most studies in Asians, where the alleles T and C of rs1048661 and rs2165241, respectively, are the risk alleles”.

6. Pag 4, Introduction: Please, change the sentence: “However, no significant association between XFS/XFG and SNP rs1048661 was observed” to “However, in contrast to what was observed in most Caucasian populations, no significant association between XFS/XFG and SNP rs1048661 was observed”.

7. Page 6, Study characteristics: Figure 1 has not been corrected according to the text.

Original figure 1 has been included in the revised manuscript

“Our initial literature search returned 197 articles. Upon browsing the titles and abstracts, 111 articles were excluded, leaving 86 articles that underwent full-text review”.

Titles and abstracts were browsed in the first step, not in the second as is shown in the figure 1.

8. Please, review the number of records excluded because they involved other LOXL1 gene polymorphisms. 32 in the text and 31 in the figure 1.

Review also the numbers in the text: “Although five articles [10,30-33] had been analyzed in a previous meta-analysis [34], we excluded them because three articles [10,30,31] did not achieve HWE in the control group, and two articles [32-33] reported the relationship between LOXL1 gene polymorphisms and primary open-angle glaucoma”. The numbers in the text must match those in the figure 1.

9. Figure 5 appears in the text before figure 4.

10. Figure 7 appears in the text before figure 6.

11. Discussion, pag 20: “Based on previous studies and the findings from the present meta-analysis, we hypothesized that the LOXL1 gene polymorphisms may upregulate LOXL1 production and that the high serum concentration of LOXL1 may contribute to XFS/XFG susceptibility [75,76]”.

Neither the present study nor the references provided by the authors support this hypothesis.

The works of Greene et al. (75) and Want et al. (76) have not studied the relationship between LOXL1 variants and dysregulation of LOXL1 production. According to Greene et al., (75) LOXL1 expression is downregulated in XFG due to DNA methylation and LOXL1 promoter methylation, while Want et al. (76) associate XFS with dysfunction in autophagy. In these studies there is no mention of LOXL1 variants

On the other hand, the present study has not analyzed the production of LOXL1 and its relationship with LOXL1 variants.

No indication has been provided to support that LOXL1 gene variants may upregulate LOXL1 production.

7. PLOS authors have the option to publish the peer review history of their article (what does this mean?). If published, this will include your full peer review and any attached files.

Reviewer #1: No

---

## [Author Response · Author response to Decision Letter 1]

20 Mar 2021

Reply: The authors would like to thank the Reviewers and the Editor for their time in evaluating our study and for making useful suggestions. We believe that we have addressed all of the comments below, but we are open to any additional ones. 

Reply to Editors： 

1. Any changes to the reference list should be mentioned in the rebuttal letter that accompanies your revised manuscript. 

Reply: We apologize for the mistake in quoting the two references [75,76] because we misunderstand the two references’ meaning. So, we removed these two references. 

Discussion, pag 20: “Based on previous studies and the findings from the present meta-analysis, we hypothesized that the LOXL1 gene polymorphisms may upregulate LOXL1 production and that the high serum concentration of LOXL1 may contribute to XFS/XFG susceptibility [75,76]”. Neither the present study nor the references provided by the authors support this hypothesis.The works of Greene et al. (75) and Want et al. (76) have not studied the relationship between LOXL1 variants and dysregulation of LOXL1 production. According to Greene et al., (75) LOXL1 expression is downregulated in XFG due to DNA methylation and LOXL1 promoter methylation, while Want et al. (76) associate XFS with dysfunction in autophagy. In these studies, there is no mention of LOXL1 variants. On the other hand, the present study has not analyzed the production of LOXL1 and its relationship with LOXL1 variants.No indication has been provided to support that LOXL1 gene variants may upregulate LOXL1 production. 

Reply to Reviewers： 

Reply: We thank the reviewers’ comments and excellent suggestions. We have revised these mistakes and inaccuracies in the text as you pointed out, rewritten some references and revised some parts of the text. Reply to comments and suggestions of reviewers as follows:

1. Pag 1, Abstract Background, line 1: LOXL1 (do not use italics when referring to protein)

Reply: We have corrected the text of LOXL1.

2. Pag 1, Abstract Results: The number of controls in the sentence “In total, 5022 cases and 10400 controls were included in this meta-analysis” should be corrected.Table 1 show the number of XFS and XFG cases and controls in each study. Some studies report the results for XFS and XFG separately, in these studies the control group is the same for both diseases, but it was counted in duplicate. For example, in the case of the study of Panday et al., there is only a group control with 61 individuals.

Reply: In total, 5022 cases and 8962 controls were included in this meta-analysis. The total number of participants has been corrected in discussion too

3. Pag 2, Abstract results: Delete the word “only” in the sentence: “In addition, only rs1048661 and rs3825942 correlated with XFS/XFG susceptibility in Africans.The SNP rs2165241 was not analyzed in Africans. Authors do not know if this SNP is associated with XFS/XFG.

Reply: We have deleted the word “only” in the sentence.

4. Pag 3, Introduction: in the sentence “…as well as in most studies carried out in Caucasians [16,17]” delete reference 17, and add references of other studies in Caucasians

Reply: We have replaced the reference 17 to the reference 13

5. Pag 3-4, Introduction: Delete the word “where” in the sentence: “However, in most studies in Asians, where the alleles T and C of rs1048661 and rs2165241, respectively, are the risk alleles”.

Reply: We have deleted the word “where” in the sentence

6. Pag 4, Introduction: Please, change the sentence: “However, no significant association between XFS/XFG and SNP rs1048661 was observed” to “However, in contrast to what was observed in most Caucasian populations, no significant association between XFS/XFG and SNP rs1048661 was observed”.

Reply: We have changed the sentence: “However, no significant association between XFS/XFG and SNP rs1048661 was observed” to “However, in contrast to what was observed in most Caucasian populations, no significant association between XFS/XFG and SNP rs1048661 was observed”.

7. Page 6, Study characteristics: Figure 1 has not been corrected according to the text.

Original figure 1 has been included in the revised manuscript

“Our initial literature search returned 197 articles. Upon browsing the titles and abstracts, 111 articles were excluded, leaving 86 articles that underwent full-text review”.

Titles and abstracts were browsed in the first step, not in the second as is shown in the figure 1.

Reply: The 197 articles were retrieved without screening, so, titles and abstracts were browsed in the second step.

8. Please, review the number of records excluded because they involved other LOXL1 gene polymorphisms. 32 in the text and 31 in the figure 1.

Review also the numbers in the text: “Although five articles [10,30-33] had been analyzed in a previous meta-analysis [34], we excluded them because three articles [10,30,31] did not achieve HWE in the control group, and two articles [32-33] reported the relationship between LOXL1 gene polymorphisms and primary open-angle glaucoma”. The numbers in the text must match those in the figure 1.

Reply: There are some vague descriptions in the article, and the numbers in the text are match those in Figure 1 indeed. So, we have revised some parts of the text to match with the Fig 1.

9. Figure 5 appears in the text before figure 4.

Reply: We have revised it.

10. Figure 7 appears in the text before figure 6.

Reply: We have revised it.

11. Discussion, pag 20: “Based on previous studies and the findings from the present meta-analysis, we hypothesized that the LOXL1 gene polymorphisms may upregulate LOXL1 production and that the high serum concentration of LOXL1 may contribute to XFS/XFG susceptibility [75,76]”.

Neither the present study nor the references provided by the authors support this hypothesis.

The works of Greene et al. (75) and Want et al. (76) have not studied the relationship between LOXL1 variants and dysregulation of LOXL1 production. According to Greene et al., (75) LOXL1 expression is downregulated in XFG due to DNA methylation and LOXL1 promoter methylation, while Want et al. (76) associate XFS with dysfunction in autophagy. In these studies there is no mention of LOXL1 variants

On the other hand, the present study has not analyzed the production of LOXL1 and its relationship with LOXL1 variants.

No indication has been provided to support that LOXL1 gene variants may upregulate LOXL1 production.

Reply: We apologize for the mistake in quoting the two references [75,76] because we misunderstand the two references’ meaning. So, we removed these two references. 

Sharma et al. [59] reported that the coding variants rs1048861 and rs3825942 may alter protein function and binding, wherein molecular modeling displayed that positions 141(rs1048661) and 153(rs3825942) of the LOXL1 protein are likely surface residues and hence possible recognition sites for protein-protein interactions. Alterations at these residues might change the capacity of LOXL1 to bind other proteins related to its cleavage as well as processing. Nevertheless, the difference in processing of the LOXL1 protein variants detected in their research does not completely interpret susceptibility to XFS/XFG among carriers of these variants as each of the variants confer the XFS/XFG risk in various ethnicities. The detailed mechanism whereby LOXL1 gene polymorphisms lead to the XFS/XFG, remains poorly understood. Therefore, Further studies are required to elucidate the mechanism on how the LOXL1 gene polymorphisms impact the occurrence and development of XFS/XFG.

---

## [Decision Letter · Decision Letter 2]

8 Apr 2021

PONE-D-20-37687R2

LOXL1 gene polymorphisms are associated with exfoliation syndrome/exfoliation glaucoma risk: an updated meta-analysis

PLOS ONE

Dear Dr. Jie,

Thank you for submitting your manuscript to PLOS ONE. After careful consideration, we feel that it has merit but does not fully meet PLOS ONE’s publication criteria as it currently stands. Therefore, we invite you to submit a revised version of the manuscript that addresses the points raised during the review process.

The reviewer has identified a few minor writing errors that need to be corrected, and the journal does not perform copy editing. When these have been corrected, your manuscript will be accepted. Thank you for submitting your work to PLoS One. 

Please submit your revised manuscript within two weeks. If you will need more time than this to complete your revisions, please reply to this message or contact the journal office at plosone@plos.org. Please include the following items when submitting your revised manuscript:

We look forward to receiving your revised manuscript.

Kind regards,

Philip C. Trackman, Ph.D.

Academic Editor

PLOS ONE

Journal Requirements:

Reviewers' comments:

Reviewer's Responses to Questions

**Comments to the Author**

1. If the authors have adequately addressed your comments raised in a previous round of review and you feel that this manuscript is now acceptable for publication, you may indicate that here to bypass the “Comments to the Author” section, enter your conflict of interest statement in the “Confidential to Editor” section, and submit your "Accept" recommendation.

Reviewer #1: All comments have been addressed

2. Is the manuscript technically sound, and do the data support the conclusions?

Reviewer #1: (No Response)

3. Has the statistical analysis been performed appropriately and rigorously? 

Reviewer #1: (No Response)

4. Have the authors made all data underlying the findings in their manuscript fully available?

Reviewer #1: (No Response)

5. Is the manuscript presented in an intelligible fashion and written in standard English?

Reviewer #1: (No Response)

6. Review Comments to the Author

Reviewer #1: This manuscript is acceptable for publication. However, some misprints should be corrected

Abstract, Background: “… in the gene encoding the LOXL1…” should be “… in the gene encoding LOXL1…”

Discussion, Pag 18: “… the allele T of rs2165241 had a potential protective effect on XFS/XFGS in Asians.” XFG not XFGS

Discussion, Pag 19: “However, we also found that LOXL1 the G allele of rs1048661..." should be “However, we also found that the G allele of rs1048661 ..."

Discussion, Pag 19: “Additionally, there was a significant association between LOXL1 gene polymorphisms and susceptibility to various disease types. These results affirmed the association between LOXL1 gene polymorphisms and XFS and XFG.” LOXL1 in italics when referring to the gene

7. PLOS authors have the option to publish the peer review history of their article (what does this mean?). If published, this will include your full peer review and any attached files.

Reviewer #1: No

---

## [Author Response · Author response to Decision Letter 2]

8 Apr 2021

Reply: The authors would like to thank the Reviewers and the Editor for their time in evaluating our study and for making useful suggestions. We believe that we have addressed all of the comments below, but we are open to any additional ones. 

Reply to Reviewers： 

Reply: We thank the reviewers’ comments and excellent suggestions. We have revised these mistakes as you pointed out. Reply to comments and suggestions of reviewers as follows:

1. Abstract, Background: “… in the gene encoding the LOXL1…” should be “… in the gene encoding LOXL1…”

Reply: We have revised the mistake in page 1.

2. Discussion, Pag 18: “… the allele T of rs2165241 had a potential protective effect on XFS/XFGS in Asians.” XFG not XFGS

Reply: We have revised the mistake in page 18.

3. Discussion, Pag 19: “However, we also found that LOXL1 the G allele of rs1048661..." should be “However, we also found that the G allele of rs1048661 ..."

Reply: We have revised the mistake in page 19.

4.Discussion, Pag 19: “Additionally, there was a significant association between LOXL1 gene polymorphisms and susceptibility to various disease types. These results affirmed the association between LOXL1 gene polymorphisms and XFS and XFG.” LOXL1 in italics when referring to the gene

Reply: We have corrected the text of LOXL1 in page 19.

---

## [Editor Report · Decision Letter 3]

14 Apr 2021

LOXL1 gene polymorphisms are associated with exfoliation syndrome/exfoliation glaucoma risk: an updated meta-analysis

PONE-D-20-37687R3

Dear Dr. Jie,

We’re pleased to inform you that your manuscript has been judged scientifically suitable for publication and will be formally accepted for publication once it meets all outstanding technical requirements.

Kind regards,

Philip C. Trackman, Ph.D.

Academic Editor

PLOS ONE
---

## [Editor Report · Acceptance letter]

16 Apr 2021

PONE-D-20-37687R3 

*LOXL1* gene polymorphisms are associated with exfoliation syndrome/exfoliation glaucoma risk: An updated meta-analysis 

Dear Dr. He:

I'm pleased to inform you that your manuscript has been deemed suitable for publication in PLOS ONE. Congratulations! Your manuscript is now with our production department. 

Kind regards, 

on behalf of

Dr. Philip C. Trackman 

Academic Editor

PLOS ONE